# Strong and Precise Modulation of Human Percepts via Robustified ANNs

**Guy Gaziv**[*]    **Michael J. Lee**[*]    **James J. DiCarlo**
McGovern Institute for Brain Research, Dept. of Brain and Cognitive Sciences
Massachusetts Institute of Technology

## Abstract

The visual object category reports of artificial neural networks (ANNs) are notoriously sensitive to tiny, adversarial image perturbations. Because human category reports (aka human percepts) are thought to be insensitive to those same small-norm perturbations – and locally stable in general – this argues that ANNs are incomplete scientific models of human visual perception. Consistent with this, we show that when small-norm image perturbations are generated by standard ANN models, human object category percepts are indeed highly stable. However, in this very same "human-presumed-stable" regime, we find that *robustified* ANNs reliably discover low-norm image perturbations that strongly disrupt human percepts. These previously undetectable human perceptual disruptions are massive in amplitude, approaching the same level of sensitivity seen in robustified ANNs. Further, we show that robustified ANNs support precise perceptual state *interventions*: they guide the construction of low-norm image perturbations that strongly alter human category percepts toward specific prescribed percepts. In sum, these contemporary models of biological visual processing are now accurate enough to guide strong and precise interventions on human perception.

*Code*    *Webpage*

## 1 Introduction

Based on empirical alignment, some fully-trained artificial neural networks (ANNs) are the current leading scientific models of the integrated mechanisms of the adult primate visual ventral stream – in particular, its multi-level neural computations and some of the visually-guided behaviors it supports [1–10]. However, individual ANNs are also notoriously susceptible to *adversarial attack*: Adding of a tiny (e.g., ultra-low norm $\ell_2$-norm) pixel perturbation to the model's input, which is optimized to disrupt the model's categorization of the original image [11–13]. Because human object category perception has been shown to be only weakly sensitive to those same small-norm pixel perturbations [14–16], if at all, this argues against those ANNs as scientific models of human visual perception [17, 18].

Moreover, human object category perception is empirically robust to random image perturbations of sufficiently low-norm, here estimated to be $||\delta||_2 \leq 30$ (see Methods). The corresponding assumption is that humans are robust to *any* perturbation with low norm (Fig 1a, but see [19, 20] for special cases). Our primary goal here was to re-examine this prevailing assumption, and to systematically compare human and ANN model categorization behavior in this low-norm pixel perturbation regime.

We were motivated by three recent observations: First, much recent work has been aimed at addressing the adversarial sensitivity of ANNs [21], including *adversarial training* [22] – a training scheme in which insensitivity to low-norm pixel perturbations is *explicitly imposed* by generating adversarial examples during model optimization (a procedure we refer to here as "robustification"). Intriguingly, the resultant *robustified* ANNs are not only less susceptible to adversarial attacks on new clean

---

*Equal contribution

37th Conference on Neural Information Processing Systems (NeurIPS 2023).

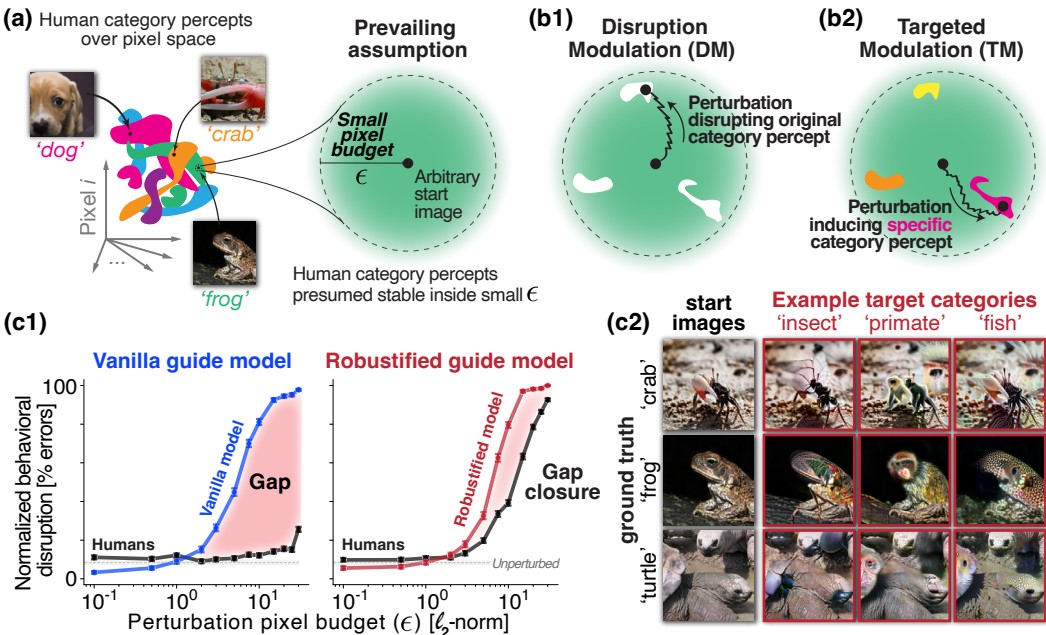

Figure 1: **Robustified models discover low-norm image perturbations that strongly modulate human category percepts.** *The prevailing assumption: Human object category percepts have complicated topology in pixel space, but are robust (i.e., stable) inside a low pixel budget envelope around most natural images (**a**). Guided by robustified ANNs, we discovered image perturbations within that envelope that strongly disrupt (**b1**) or precisely change (**b2**) human category percepts. (**c1**) Image category disruption rates of humans (black) and models on the same modulated images, replicating a large gap in behavioral alignment for a non-robust "vanilla" model, which is largely closed by adversarial robustification. Model curves show reports by "surrogate" model of the Guide Model type (i.e., "gray-box"). (**c2**) Robust models allow for precise Targeted Modulation (TM) of human behavior in the "human-presumed-insensitive" pixel budget regime. Shown are TM examples by $\ell_2$ pixel budget of 30.0 in the low pixel budget regime, demonstrating that arbitrary source images can be locally modulated toward arbitrary target human category percepts. The presented robust model in all panels is adversarially-trained at $\ell_2$ pixel budget of 3.0. Error bars, 95% CI over source images & human raters.*

images [22, 23], but, unlike vanilla ANN attacks, those attacks introspectively seem to engage human perception [24–28]. Second, when tested on clean natural images, robustified ANNs have internal "neural" representations that are more closely aligned with the corresponding neural population representations along the primate ventral stream, relative to vanilla ANNs [29, 30]. Third, the adversarial sensitivity of the responses of individual "neurons" in deep layers of robustified ANNs has been reported to be comparable with the sensitivity of the responses of individual biological neural sites at the corresponding layer of the primate visual ventral stream [31]. In addition, the robustified ANNs were found to reliably predict [ultra-]low-norm perturbation *directions* in pixel space for which biological neural sites turned out to indeed exhibit unexpectedly high sensitivity.

Taken together, these observations suggested the possibility that robustified ANNs may have closed the human-to-model behavioral gap in low-norm perturbation sensitivity. If so, then because robustified ANNs still exhibit ubiquitous, dramatic category report changes with precisely targeted low-norm pixel perturbations, then humans must *also* exhibit such not-yet-detected, ubiquitous, massive "bugs" in their visual processing. And those bugs could be utilized as image-dependent "features" that may give rise to very strong, yet very efficient (i.e., low pixel budget) human perceptual modulation.

We therefore asked: **Q1) have robustified ANN models closed the model-human gap in adversarial sensitivity?** And, **Q2) can arbitrary human percepts be specifically induced in the low pixel budget regime**, where human percepts are believed to by highly robust? We asked these two questions in the context of nine-way visual categorization tasks that are performed both by ANN models (all parameters frozen) and by human observers: a test image is presented and one of nine category reports is made (by models and by humans). To create each such test image, a frozen ANN model is utilized by a standard image generation algorithm as a "guide" in pixel space (dimensionality D = 150,528; see Methods) to construct a low-norm pixel perturbation (length D) relative to an arbitrary start image for a prescribed "perceptual" goal. For **Q1)** the prescribed goal is: "Perturb this start image to disrupt the model's currently category judgement." And, for **Q2)** the prescribed

goal is: "Perturb this start image to induce a prescribed category judgement [X]." We use different guide models to generate these perturbations, and we then measure the effect (if any) of such image perturbations on the categorization behavior of: (i) the original ANN guide model (aka "white-box"), (ii) a surrogate ANN model from the same family as the guide model (aka "gray-box"/transfer test), and (iii) a pool of human observers at natural viewing times.

Our main contributions are as follows:
• We provide evidence for the existence of low-norm image perturbations of arbitrary natural images that strongly disrupt human categorization behavior.
• We provide empirical evidence for the existence of local (i.e., low-norm) perturbations in image space that support changes from the category percept induced in a human subject by an arbitrary start image to semantically very different perceptual categories.
• We show that some robustified ANN models have largely, but not completely, closed the model-to-human behavioral correspondence gap in low-norm perturbation sensitivity.
• We show that robustified ANN models coupled with a generator algorithm can support surprisingly precise, low-norm perceptual state modulations for humans.

## 2 Overview of approach and experiments

To study the effects of small image perturbations on human visual object categorization, we used a two-stage methodology: (i) generate small image perturbations predicted to modulate human behavior by highly-ranked models of the ventral visual stream and control models, then (ii) collect human object categorization reports in a nine-way choice task, identical to that performed by the model using Amazon Mechanical Turk surveys. This methodology allowed us to directly assess the alignment between model category judgements and (pooled) human category judgements in response to the same start images and the same perturbations of those start images.

### 2.1 Generating image perturbations predicted to modulate human behavior

We focused on two image perturbation modes conceptually illustrated in Fig 1b: (i) Disruption Modulation (DM), involving perturbations intended to induce model errors by driving the model's category judgement away from the ground-truth category label, irrespective of the alternatively-induced model category judgement (Q1 above); and (ii) Targeted Modulation (TM), involving perturbations designed to induce specific, "target" model category judgements (Q2 above). These two modes are also referred to as untargeted and targeted attacks, respectively [22, 32, 18]. Image perturbations were created by one of the four model families we considered here, without any guidance from human experimenters and without guidance from human subjects data.

To have a tractable perceptual category report set that avoids the problem of humans being unfamiliar with the 1000 fine-grained ImageNet classes, we focused on a subset of ImageNet that has been mapped to a basic set of nine classes, called "Restricted ImageNet" [32]. Using this nine-category image set, we randomly sampled starting images (aka "clean" images) from each class and, for each such starting image, produced a DM-perturbation image and eight TM-perturbation images (one for each of the eight alternative possible categories). Each such perturbation image was required to be within the pixel "vicinity" of the starting image by imposing a pre-chosen upper limit on the $\ell_2$-norm of the perturbation (aka "pixel budget" $\epsilon$, Fig 1a). To systematically test perturbation efficiency, we repeated this procedure for a wide range of pre-chosen pixel budget limits, focusing on the "human-presumed-stable" low-norm pixel budget regime ($\epsilon \leq 30$; see Sec 1).

For Guide Models (GM) used to direct the construction of image perturbations, we focused on the ResNet50 ImageNet-pretrained model family [33]. These ANNs are among the currently leading models of primate ventral visual processing stream and its supported behaviors [34, 2, 31], and particularly so in their more recent adversarially-trained, "robustified" version [22, 32]. We thus considered four families of ResNet50 models, which differ only in their robustification level, starting from the non-robust, "vanilla" variant, through those commonly employed ($\ell_2$ pixel budget at training of 3.0) [32], to new models which we trained for other robustification levels . Notably, we denote this *adversarial training budget* by $\varepsilon$, to be distinguished from the perturbation budget, $\epsilon$, which applies to any image-perturbation method.

For a fair quantitative comparison of model category judgements and human category judgements we addressed the "gray-box" setting [12, 11, 13], where we ask *"surrogate" models* – a model from

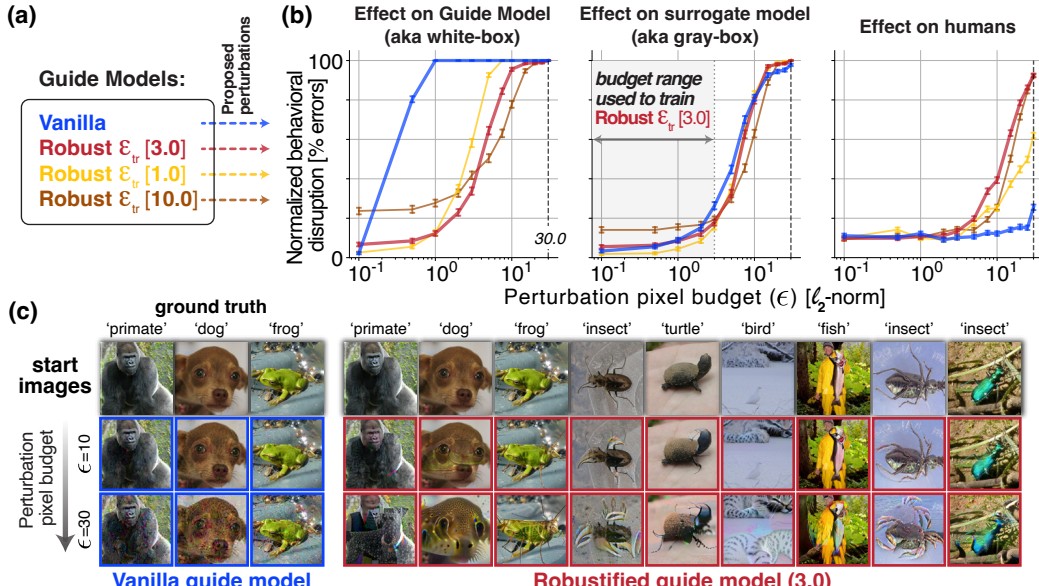

Figure 2: **Low-norm image perturbations discovered by robustified models strongly disrupt human category judgements.** *(a) The Guide Models used for Disruption Modulation (DM) image generation. (b) Disruption rates of humans and models. All panels share the same set of start images, and the four sets of perturbed images (one for each of the four Guide Models, color-coded). Y-axis shows the categorization "error" rate with respect to the category label of the (clean) start image. Human curves show average across subjects and trial repeats per image, with lapse-rate correction applied subject-wise. "White-box" refers to reports made by the Guide Model itself. "Gray-box" refers to reports by a second seed "surrogate" model of Guide Model type. Error bars, 95% CI by bootstrap over images & subjects. (b) Example of modulated images generated by robust model [3.0] (in red) and, in contrast, by vanilla model (in blue) in the low pixel budget regime.*

the Guide Model's family, but which is initialized and trained using a different seed – to report its category judgements of the perturbed images (i.e., within family transfer test).

## 2.2 Measuring human object category percepts

We used Mechanical Turk for large-scale behavioral experiments of nine-way image categorization. Randomly-selected human raters provided their trial-by-trial category reports via a sequence of trials. In each trial a single test image – randomly-chosen from the full set of TM or DM perturbations images – was presented at the center of gaze for a fixed duration of 200ms, after which the rater was shown a category choice screen with nine category options ('dog', 'cat', 'frog', 'turtle', 'bird', 'primate', 'fish', 'crab', 'insect'). No pre-mask or post-mask was used. Raters were asked to report the category that best describes the presented image.

## 3 Results

### 3.1 Disruption Modulation results

Fig 2 shows the results of the Disruption Modulation (DM) experiment. The DM image generation Guide Models include a non-robust (vanilla) model and three versions of robustified models, trained to different robustification levels, indicated by their perturbation pixel budget at adversarial training, namely, 1.0, 3.0, and 10.0. Each panel corresponds to measured category report "errors" on a different system: the guide model itself, a guide-surrogate model, and on humans. Note that "errors" here are defined with respect to the ground truth label of the start (unperturbed) image.

Examination of the effect of image perturbations on the Guide Model that designed them (Fig 2b, left) confirms previous reports that robustification strongly reduces white-box perturbation sensitivity near novel start images, as previously reported [11–13]. And the human DM results confirms previous reports [14–16] that human category percepts are only very weakly sensitive to low-norm image perturbations designed by vanilla ANNs (Fig 2b right, blue line).

In contrast, we report this novel finding: **human percepts are massively disrupted by low-norm image perturbations discovered by robustified ANNs** trained at an $\ell_2$ pixel budget of 3.0 (red line). At an image perturbation budget of 30, ∼90% of human category reports are no longer the category of the start image. Visual inspection of some example test images (Fig 2c) allows introspection about the perceptual differences between image perturbations generated by vanilla guide models vs. those generated by robustified guide models. Notably, we report these strong disruptions in human category percepts when perturbing at budgets well above the budget used for model robustification, yet still well-below the typical pairwise natural image distance regime (see Supplementary Material, Fig 1). This result was not a priori guaranteed.

We sought to determine whether human perceptual disruptions are quantitatively aligned with the predictions of robustified ANNs. Because we cannot perform white box attacks on the human visual system, the fairest assessment of this is via comparison of effects on humans with effects on surrogate models. To facilitate this comparison, we re-plot the results of the robustified ANN model (level 3.0, red) on the same panel as the human data (Fig 1c1, black line). Those plots demonstrate: (i) the originally motivating qualitative "gap" between the very high sensitivity of vanilla (surrogate) ANNs and the almost complete lack of human perceptual sensitivity to the same perturbations, and (ii) that some robustified ANNs have dramatically closed that gap. Nevertheless, some human-model misalignment still exists, with humans typically requiring about twice the perturbation strength to produce a quantitatively matched level of perceptual disruption as that particular ANN model family (red vs. black in Fig 1c1 right).

We also noted that robustified models show stronger "white-box"-to-"gray-box" alignment. Namely, image perturbations driven by robustified models not only transfer effectively to humans but also exhibit stronger alignment to surrogates of their own model class, compared to vanilla models (compare Fig 2b left vs. middle graphs).

We further assessed the impact of robustification training level on our findings. We found that models, which were adversarially trained at reduced allowable budget of 1.0, were less human-aligned (cf. Fig 2b right, yellow vs. red lines). Increasing the allowable budget at training to 10.0 did not result in a significant qualitative difference in model-human alignment. However, it did lead to less effective human category disruptions per budget level (cf. Fig 2b right, brown vs. red lines).

In follow up experiments, we found these human perceptual disruption effects to be largely unaffected by test image viewing times in the natural viewing time range of 100-800ms (see Supplementary Material). In summary, our results demonstrate that robustified models reveal at least a subset of behaviorally meaningful image perturbations in the low pixel budget regime.

### 3.2 Targeted Modulation results

Fig 3 shows the results of the Targeted Modulation experiments – tests of each Guide Model's ability to shift human percepts *toward prescribed target categories* from arbitrary start images (see Sec 2.1). We first randomly selected start images from all nine object categories and perturbed each source image toward all different-class targets. We used several Guide Models: the non-robust (vanilla) model, three variations of robustified models with different levels of robustification, and an image interpolation baseline approach using randomly selected images from the intended target categories. We experimented with a range of $\ell_2$ pixel budgets in the low pixel budget regime, using the same source images across all conditions. Fig 3a provides examples of images generated by the $\varepsilon_{tr}[3.0]$ robustified model at a 30.0 pixel budget in the low budget regime.

We found that robustified models trained with an $\ell_2$ pixel budget of 3.0 and 10.0 give rise to perturbations that are reliably reported by humans as the target category (Fig 3b). These models discovered specific image modifications that resulted in a 60% probability of humans choosing the prescribed target category at an $\ell_2$ pixel budget of 30 (chance: ∼11%). In comparison, the interpolation approach remained at baseline level throughout the low pixel budget regime. Consistent with the findings in the Disruption Modulation experiment, we found that perturbations made by vanilla model had no targeted effect on human category percepts.

The *specificity* of our Targeted Modulation is further demonstrated in Fig 4a. This panel shows examples of successful TM toward 'dog' and 'cat' for a robustified model ($\varepsilon_{tr}[3.0]$), and other baseline methods. The precise control of a model is shown in confusion matrices in Fig 4b. They further highlight the strong specificity of TM for the robust model at $\ell_2$ pixel budget of 30 relative to a image interpolation baseline approach.

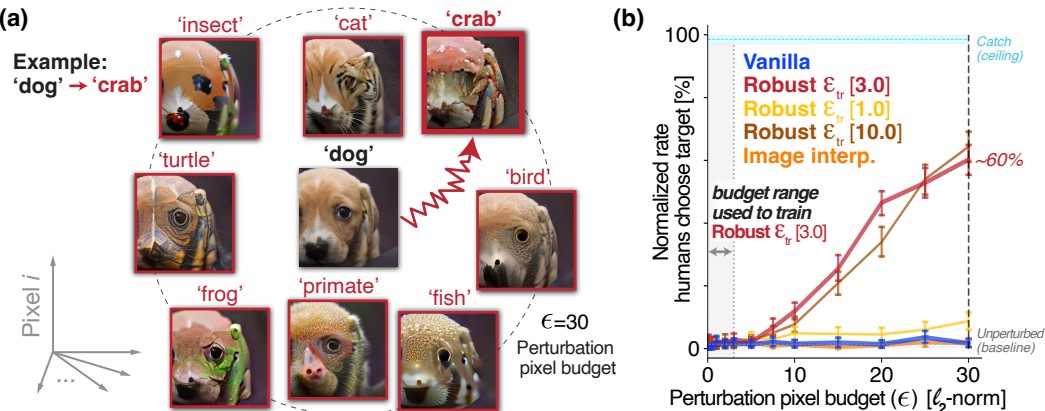

Figure 3: **Robustified models enable budget-efficient precise targeted behavioral modulation.**
*(a) Targeted Modulation (TM) example: A 'dog' start image is here independently modulated toward each of eight alternative target categories by Robust $\varepsilon_{tr}[3.0]$ at an $\ell_2$ pixel budget of 30.0. (b) Efficacy of Guide Models in modulating human reports toward specific, prescribed target categories (TM). Images were drawn from all nine categories and each image was independently perturbed towards each of the eight alternative possible target categories (as illustrated in (a)). Curves show the lapse-normalized probability of human choosing the prescribed target category. Baseline marks refer to unperturbed images, indicating human rates of reporting the target category of the source image ('Catch'). Error bars, 95% CI by bootstrap over source images, target categories, and subjects. Robust models trained with $\ell_2$ pixel budget 3.0 and 10.0 gave rise to the most budget-efficient behavior modulation.*

***Dependence on start images & target category percepts.*** To test the generality of these behavior modulation findings across different distributions of the start images, we conducted similar experiments (all at pixel budget 30) using source images from (i) Out Of Distribution (OOD) images of the same categories ('frog', etc.), (ii) Arbitrary Natural Images (ANI), collected from the internet and social media, and (iii) Uniform Noise Images (UNI). We found that the TM effect magnitude remained comparable even when start images were drawn from these different distributions. For In-Distribution (ID), OOD, ANI, and UNI we achieved normalized target category induction rates of of 60%, 51%, 55%, and 70%, respectively (Fig 4c). Notably, TM often fails to produce convincing images for some target classes when using UNI source images (see Supplementary Material).

To further test the generality of our approach, we conducted experiments using *new, arbitrary target categories*. We selected nine target classes from an external animal image dataset, using 60 randomly selected images to represent each class. We modified our TM method to optimize towards the centroid feature representation of the target class, obviating the need to train a new classifier for the new class space (see Sec 4). We found high TM specificity also under this setting (Fig 4d), with lower TM scores on average compared to previous settings. This is despite the challenges posed here, which include, in addition to using new target classes, the presence of source-image distribution shifts (i.e., OOD, ANI; For UNI see Supplementary Material).

***Induction of composite target category percepts*** We next sought to test whether model-guided perturbations could move human percepts toward pairs of object categories. Because subjects were only allowed to report one of nine categories, a perfect success in this test would be inducing humans to report each of the two prescribed target categories ∼50% of the time. Fig 4e shows results for these "two-composite" experiments. These results indicate that, while not perfect, even under this unusual setting, robustified models often discover highly creative ways to combine multi-class features *in a single image* that are human-recognizable. Visual inspection of the modulated images further highlights that the multi-class modulation cannot be trivially attributed to tiling the relevant class features across different spatial locations – The budget constraint guides the model to find clever ways *to combine* rather than "just add". Notably, scaling to 3-composite directions under the restrictive budget of 30 makes TM highly challenging, and more sensitive to the specific start images (see Supplementary Material).

***Optimal robustification level.*** We defined the Model TM Efficiency (MTME) to be the budget at which a modulatory effect size exceeding 10% probability of choosing the target is observed. To this end, we interpolated the curves from Fig 3 uniformly to 100 points in $\ell_2$ pixel budget range of $[0, 30]$, and identified the first budget at which the threshold is exceeded. We rank the models by

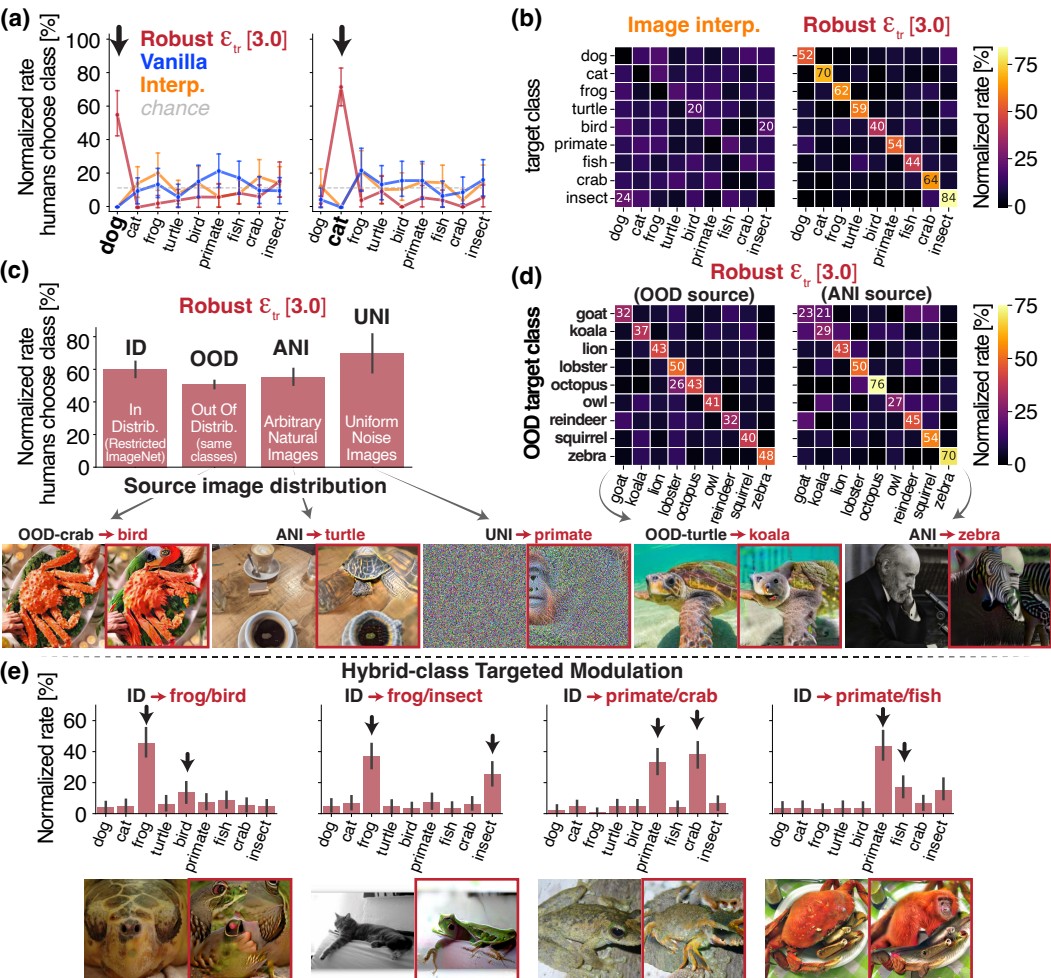

Figure 4: **Robustified models achieve precise Targeted Modulation (TM) from any image start point to any target direction.** *(a, b) Behavioral modulation results obtained when using a robust model to drive toward all possible cardinal class directions (i.e., 1-hot). (c, d) Analysis of distribution dependence of source images and target directions, generalizing our precise targeted modulation finding beyond Restricted ImageNet. (e) Targeted Modulation toward hybrid directions of two classes. All plots and images are due to robust model ($\varepsilon_{tr}[3.0]$) that perturbs natural source images at an $\ell_2$ pixel budget of 30.0. All panels show directly comparable numbers. Error bars, 95% CI over source images & human subjects.*

ascending efficiency order (marking MTME score, lower is better): Rank 5) Vanilla, which never reaches the 10% threshold within the considered range, Rank 4) Robusified $\varepsilon_{tr}[1.0]$ (196), Rank 3) Image interpolation (47), Rank 2) Robusified $\varepsilon_{tr}[10.0]$ (11), and the best, Rank 1) Robusified $\varepsilon_{tr}[3.0]$ (9). This suggests that there exist an optimal robustification level for strongest modulation effects. While this precise optimization is beyond the scope of this work, we highlight that the Robust $\varepsilon_{tr}[3.0]$ model is currently the most TM-efficient amongst all tested.

***Other pixel budget constraints.*** In addition to the $\ell_2$ perturbation constraint, we also explored the use of $\ell_\infty$ constraint for TM. Visual inspection of the resulting images revealed that both constraints induce comparable modulation effects, with the $\ell_2$ constraint giving rise to images that appeared slightly more visually "clean" and aesthetically pleasing. We also found that TM by $\ell_\infty$-trained models was significantly weaker compared to the $\ell_2$-trained models.

## 4 Methods

### 4.1 Distances in image space

We exclusively focused on $224 \times 224 \times 3$ RGB pixel space ($D = 150,528$), where each value ranges between 0 and 1. Thus, the maximal distance between any two images is $\|\delta\|_2^{max} \approx 388$. For

reference, we found the typical distance between ImageNet images to be $\approx 130$. We also estimated the regime in which human object perception is empirically robust to random perturbations to be $\|\delta\|_2 \leq 30$ (see Supplementary Materials; also see [35]).

## 4.2 Stimuli generation by model-guided perturbations

***Disruption Modulation (DM).*** We used the Projected Gradient Descent (PGD) attack algorithm [31, 36, 37], using an adapted version of the robustness library [32]. Given an initial input image and its ground-truth label in 1-hot representation $x$, $y$, and a classification model to logits for $C$ classes, $f : \mathbb{R}^{H \times W \times 3} \to \mathbb{R}^C$, we optimize for the perturbation, $\delta$, such to be disruptive of model classification to the ground-truth class label under a pixel budget restriction:

$$\delta = \underset{||\delta||_p < \epsilon}{\operatorname{argmax}} \mathcal{L}_{CE}\left(f\left(x + \delta\right), y\right), \tag{1}$$

where $\epsilon$ is the pixel budget, and $\mathcal{L}_{CE}$ is the cross-entropy loss. Unless otherwise mentioned, we focus on $\ell_2$-adversarially-trained models and attacks, i.e., $p = 2$. Optimization is performed in steps of Stochastic Gradient Descent (SGD), each followed by a projection step to pixel budget $\epsilon$:

$$\delta_{k+1} \leftarrow \operatorname{Proj}_{\epsilon}\left(\delta_k + \eta \nabla_{\delta} \mathcal{L}_{CE}\left(f\left(x + \delta_k\right), y\right)\right), \tag{2}$$

where $k$ denotes the step, and $\eta$ is the step-size. All original input images are resized to $256 \times 256$ (bilinear, anti-aliased) and center-cropped to $224 \times 224$ [22]. We focused on the following range of $\ell_2$-pixel budgets in the low-budget regime: $[0.1, 0.5, 1.0, 2.0, 3.0, 5.0, 7.5, 10, 15, 20, 25, 30, 40, 50]$. We set the number of PGD steps and the step-size, $(k_{steps}, \eta)$, such to match the pixel budget [32], ranging from $(200, 0.02)$ for $\epsilon = 0.1$ to $(2000, 2)$ for $\epsilon = 50$.

***Targeted Modulation (TM).*** The algorithm is identical to DM, aside from $y$ changing to be a given target class label, and the optimization solving for minimization in lieu of maximization (flipping of gradient sign). Notably, unlike DM, this modulation can be performed toward multiple target class labels from a single source image.

***Hybrid-Class Targeted Modulation.*** To modulate toward non-cardinal directions we defined random composite directions involving multiple classes. These directions were constructed as probability vectors with all-zero except for $k$ randomly selected classes, which were assigned equal probabilities. We generated four 2-composite directions: 'frog-bird', 'frog-insect', 'primate-crab', 'primate-fish'; Our objective function was based on cross-entropy similar to Eq 1, with additional logit maximization on the non-zero target classes, which drives higher usage of the allowable budget.

***Features Targeted Modulation.*** To modulate images toward a target feature representation we compiled a subset of 60 images per class and computed their mean feature representation using the GM. This defines the target classes via class-centroids. The optimization criterion is then the MSE between the predicted feature representation the target class-centroid.

***Image interpolation modulation.*** To generate baseline image perturbations using an image interpolation approach, we randomly drew images from a target class, used them for linear interpolation with the source image, and projected back the pixel budget envelope.

***Dataset.*** We sought to focus on a dataset with the following key properties: (i) Has a tractable class-space for exploration; (ii) Will mitigate confounds originating from typical-human unfamiliarity with the class labels (as is commonly the case in ImageNet); (iii) Is widely used in the adversarial robustness context. Based on these considerations, we found a partial and mapped version of ImageNet to a basic set of nine classes, termed "Restricted ImageNet", to be the most suitable choice (see Supplementary Material for class mapping) [32].

***Training robustified models.*** To adversarially-train models on ImageNet [38], we followed publicly released code [32]. Specifically, we trained ResNet50 models at $\ell_2$ pixel budgets of 1.0, 3.0, and 10.0, using PGD (steps, step-size) of (7, 0.3), (7, 0.5), (10, 1.5), respectively.

***Runtime.*** Our stimuli generation completes within 5min for a single batch of size 100 on a single A100 GPU. Adversarial-training of models completes within $\sim$8 days on 4 A100 GPUs .

### 4.3 Human behavioral measurements

We measured human behavior in a nine-way image categorization task. Our primary experimental objective was, for any given test image, to estimate the probability that humans would on average select each of the nine possible choices.

***Task paradigm.*** Human subjects performed "sessions" comprising multiple trials. Each trial had three stages: (i) First, the subject pressed a button at the center of their screen, encouraging them to center-fixate. (ii) A test image (covering approximately 6 degrees of visual angle) appeared at the center of their screen for 200ms. This was followed by 200ms of a blank gray screen. (iii) An array of nine labeled buttons (one for each of the nine reportable categories) appeared. These buttons were arranged radially, outside of the location of where the test image appeared. The locations of the buttons were randomly permuted on each trial, and no time limit was imposed. The subject was asked to select the button which most accurately described the image.

In total, there were $n = 130$ such trials in each session. Among these, there were four types of trials: (i) ***Warmup trials*** ($n = 10$), which were always presented at the beginning of the session. Test images were randomly sampled "clean" (unperturbed) images drawn from the Restricted ImageNet validation set [32]. (ii) ***Calibration trials*** ($n = 10$), included to measure the lapse rate of each human subject (see Lapse-rate correction). Stimulus images consisted of either a blue triangle or blue circle, and two randomly selected buttons on the choice screen were replaced with buttons for 'circle' and 'triangle'. (iii) ***Reference trials*** ($n = 10$), consisting of unperturbed stimulus images randomly selected from Restricted ImageNet validation set. (iv) ***Main trials*** ($n = 100$), consisting of potentially perturbed versions of source images. Trials types (ii)-(iv) were run randomly interleaved, following the warmup trials. In general, subject feedback was only delivered for unperturbed images. Upon an incorrect selection by the subject, a black "x" was displayed for 1000ms; No feedback was delivered upon a correct selection. We implemented the task paradigm using JSPsych, and presented the stimuli using the JSPsych-Psychophysics plugin.

***Human data collection.*** We used Amazon Mechanical Turk to recruit and collect behavioral measurements in human subjects, following a protocol approved by the [anonymized IRB information] The protocol has no greater than minimal risk involved for the subjects. We did not attempt to collect any personally identifying information; subjects were kept anonymous. We screened subjects by their performance on a similar "demo" object categorization task (see Supplementary Material). In total, we recruited a population of $n = 119$ subjects ($n = 43$ female; $n = 76$ male). Subjects were compensated for all time spent in this study, including in the recruitment phase. Subjects were free to opt out of the study at any time. We ensured that each image we included in our analysis had measurements from at least $n = 2$ different subjects.

***Lapse-rate correction.*** We assumed that measurements of human reports was possibly corrupted by a base "lapse rate", where a subject makes a random report independent of the presented test image's content. To account for that, we measured each subject's lapse rate, then adjusted their measurements using a formula that estimates their behavior under a lapse rate of zero (see Supplementary Material).

## 5 Conclusion

In this work, we systematically examine the prevailing assumption that human categorization is highly robust to low-norm image perturbations. Our findings challenge that assumption by establishing that the low-norm image perturbations suggested by robustified ANNs (but not vanilla ANNs) can strongly and precisely change human object category percepts. These results suggest two novel conclusions: (i) for arbitrary starting points in image space, there exists nearby "wormholes", local (i.e., low-norm) perturbations in image space that support "travel" from the current category perceptual state induced in the subject into a semantically very "distant" perceptual state, and (ii) contemporary scientific models of ventral visual processing are accurate enough to point us to those wormholes.

### 5.1 Limitations

Our findings raise many new questions: given a start image, does there *always* exist a nearby human perceptual wormhole to at least one another category? Are there nearby wormholes to *any* possible category? Which wormholes are common to all humans and which are more idiosyncratic? And do robustified ANNs *always* find the wormholes that do exist? And how often do they predict wormholes that do not actually exist? We do not yet know the answers to these questions, but the results presented

here offer some insight. For example, Fig 2b right panel, showing a $\sim$90% disruption effect, is consistent with the hypothesis that wormholes are abundant in image space. Figs 3 and 4 ($\sim$60% effect) suggests that multiple category portals are nearby, but that portals to all categories are not always nearby. Further work could produce tighter lower-bounds on such probabilities.

We emphasize that these are lower bound estimates because we are probing human vision with models that are imperfect approximations of the underlying neurobiology. Indeed, we do not claim that robustified ANN models discover *all* low-norm wormholes or that they *never* point to wormholes that do not actually exist. Notably, because we still see a residual gap in behavioral alignment (Figs 1c and 2b), other models, which are even more behaviorally aligned, must still exist. Specifically, our analyses focused on the ResNet50 model architecture, and mainly on $\ell_2$-norm budget robustified models and image perturbation type (see Sec 3.2 for other perturbation constraints). While generalizing to other architectures or image generation algorithms is not foundational to the our claims presented in this paper, those are interesting directions to explore in future work.

Furthermore, while this paper supports that adversarial training (AT) gives rise to an improved estimate of the adults state of the ventral stream and its supported behavior, we do not claim that AT is the mechanism by which robustness in humans emerges. Future work may consider alternative, more biologically plausible mechanisms (e.g., Topographic ANNs [39]) that may give rise to a comparably aligned models in terms of predictivity.

## 5.2   Societal impact

Image generation methods such as those used here have potential societal benefits, but also potential risks [40]. The scientific knowledge of brain function contributed here may present additional future risks, in that it may widen the ability of malicious actors to subtly and non-consensually disrupt the perception of humans. Building defenses against those manipulations, and orienting the use of this knowledge toward societal benefits (e.g., consensually enriching perception, improving mental health, accelerating human visual learning) are crucial considerations in future work.

## Acknowledgments

This work was partially funded by the Office of Naval Research (N00014-20-1-2589, JJD); (MURI, N00014-21-1-2801, JJD), the National Science Foundation (2124136, JJD) and the Simons Foundation (542965, JJD).

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
