# Strong and Precise Modulation of Human Percepts via Robustified ANNs

## Supplementary Material

**Pixel budget regimes**

### Human perceptual sensitivity to random perturbations

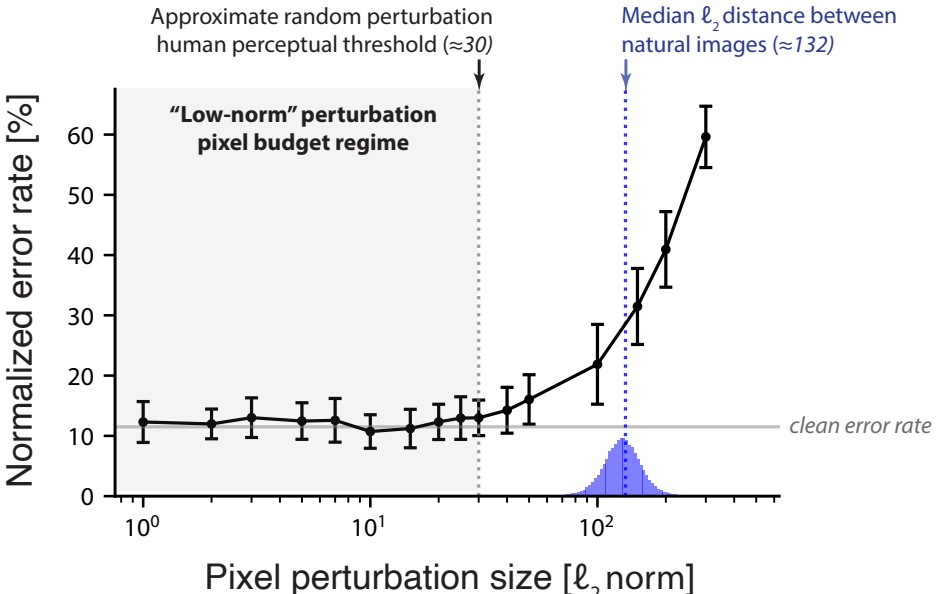

Figure 1: **Human recognition performance vs. random pixel-space perturbations.** In these tests, for a given pixel budget, the perturbations were selected to be random directions in pixel space with the same magnitude as the specified pixel budget. Specifically, perturbations were created by i) sampling from a multivariate standard normal distribution (D=150,528), ii) adjusting its norm to the current pixel budget, then iii) clipping the perturbation vector so values in the resultant perturbed image remained within the [0, 1] range. Normalized (i.e. lapse-rate corrected) human error rates (on nine-way categorization on Restricted Imagenet) are shown, as a function of the strength of random perturbations applied to the original, clean images. The Under our definition of "low budget regime" of $\epsilon \leq 30$, human error rates are nearly identical to the error rates on clean images (clean: $11.6\% \pm 2.6$; perturbed: $13.4\% \pm 3.0$). Error bars are bootsrapped SEM over images and human subjects. The empirical distribution of pairwise $\ell_2$ distances between images in the Restricted Imagenet subset are shown in the blue histogram.

**Task paradigm**

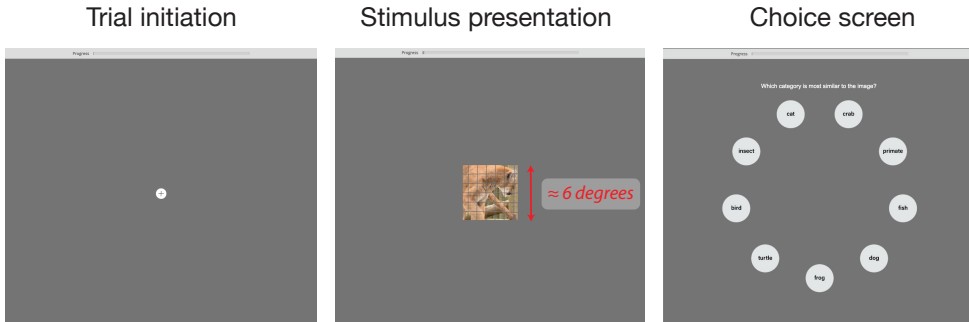

Figure 2: **Overview of task paradigm.** *Each trial consisted of three phases: trial initiation, in which the subject pressed a small white button at the center of the screen, stimulus presentation, in which a test image was briefly flashed at the center of the screen (at approximately 6 degrees of visual angle), and a choice screen consisting of 9 buttons, each labeled with a possible category report.*

**Subject screening**

To gain entry into the study, subjects were required to first perform a "demo" task consisting of 100 trials on a basic, two-way object recognition task on ImageNet [1] images, which followed the same structure as the task paradigm described in the Main Text (here, only $n = 1$ distractor was presented on each trial). We included all subjects who achieved at least 9/10 correct choices on catch trials, which were randomly interleaved throughout the session.

**Lapse-rate correction**

To account for the possibility that – unlike deterministic models – human subjects do not maintain attention to the task on 100% of the trials (referred to a "lapse-rate"), we designed catch trials, in which we assumed fully attentive subjects would achieve an error rate of zero. Thus, any empirically measured error rates above zero indicated that subject's lapse-rate, $\gamma$. The subject-wise lapse-rate corrected probability of choosing a target class thus reads, $p = \frac{\hat{p} - \frac{\gamma}{N}}{1 - \gamma}$, where $\hat{p}$ and $N$ are a subject's average measured probability on the calibration trials and the number of class options in the task (nine) respectively. Applying this correction subject-wise also allowed us to reduce the noise stemming from cross-subject baseline performance variability when estimating subject mean scores.

**Derivation**

The lapse rate is the probability that a subject will make a random choice, irrespective of the presented stimulus. We assume that this random choice is drawn from a uniform distribution; so $\frac{1}{N}$ probability for each of the $N$ class options. We call the probability they will make such a random choice $\gamma \in [0, 1]$. For any empirically measured condition, we thus have an empirically observed probability $p'$, corrupted by lapse rate $\gamma$:

$$p' = p(1 - \gamma) + \frac{\gamma}{N}. \tag{1}$$

To estimate the underlying probability, $p$, we design calibration trials, where by assumption, the subjects have an underlying perfect score, namely $p = 1$. It follows that the measured probability for those trials reads,

$$p'_{calib} = (1 - \gamma) + \frac{\gamma}{N},$$

thus the lapse-rate can be computed as,

$$\gamma = \frac{p'_{calib} - 1}{\frac{1}{N} - 1}.$$

The lapse-rate correction thus take the form,

$$p = \frac{\hat{p} - \frac{\gamma}{N}}{1 - \gamma}. \tag{2}$$

Applying this correction subject-wise also allowed us to reduce the noise stemming from cross-subject baseline performance variability when estimating subject mean scores. We refer to measures of human choice probability that are lapse-rate correct in this manner as "Normalized" (e.g., Supp. Fig 1, Main text Figs 1-4).

The typically observed lapse rates were quite low (median over subjects: 0%; mean 4.9%), indicating that not making this correction would have only a minor quantitative effect on the presented results.

**Viewing time dependence**

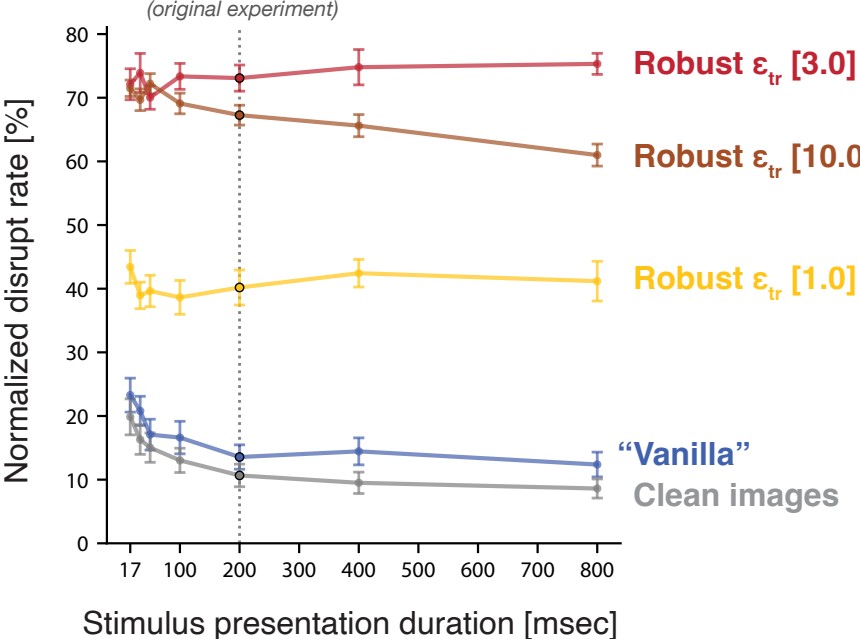

Figure 3: **Human disruption rates are largely stable across stimulus presentation times.** *We measured human disruption rates across several viewing times for a subset of Guide Model DM image perturbations ($\ell_2$ pixel budget 20.0). We found that our originally observed effects (measured at t=200 milliseconds) were replicated even at long presentation times, which permit a subject to potentially more closely analyze the image. At shorter viewing times, we observed modest or no increases in disruption rate. Errorbars are SEM (simple bootstrap over subjects and images).*

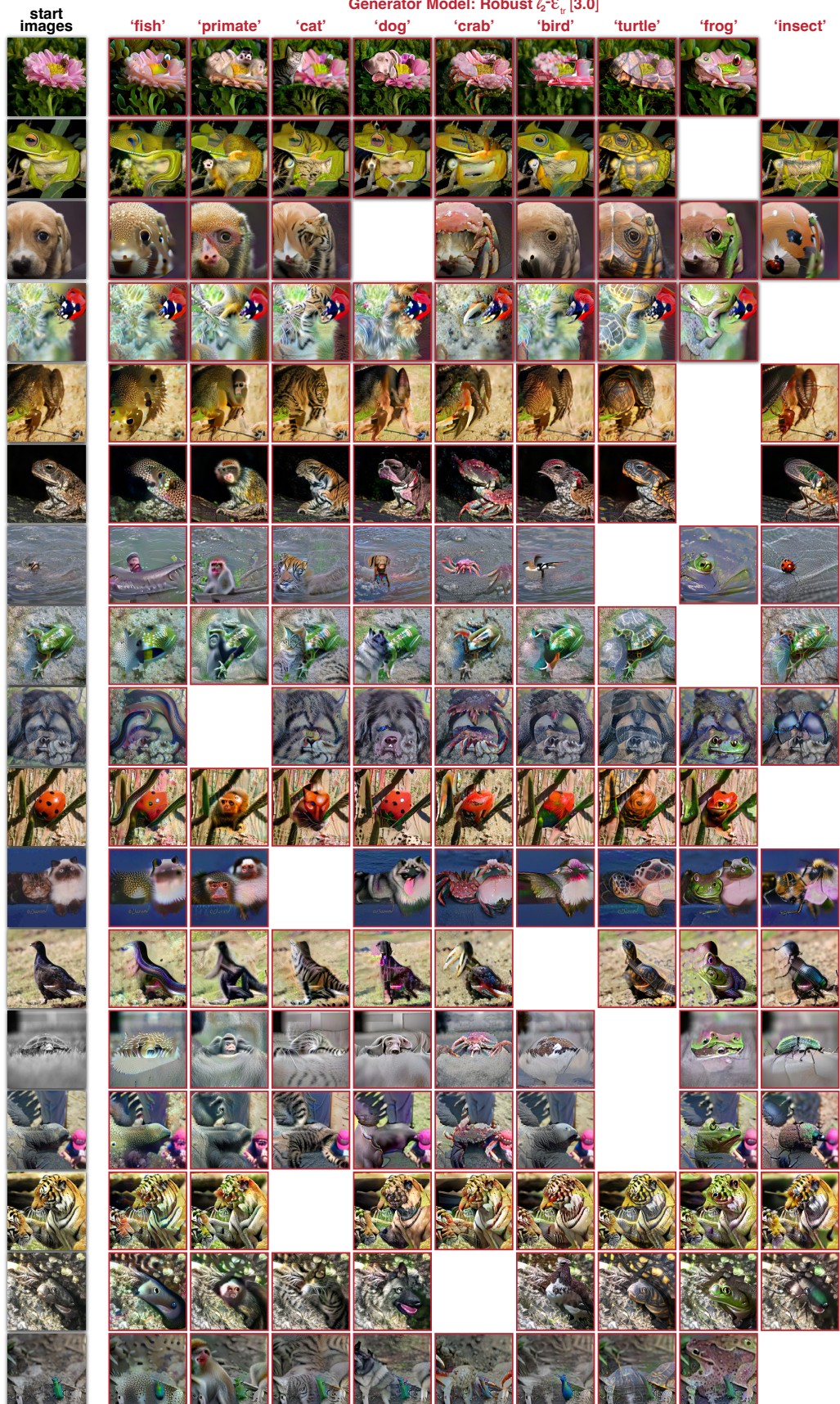

Figure 4: **Targeted Modulation examples.** *Perturbing at a pixel budget of 30. Showing randomly-selected modulation over Restricted ImageNet (see Main Text).* 4

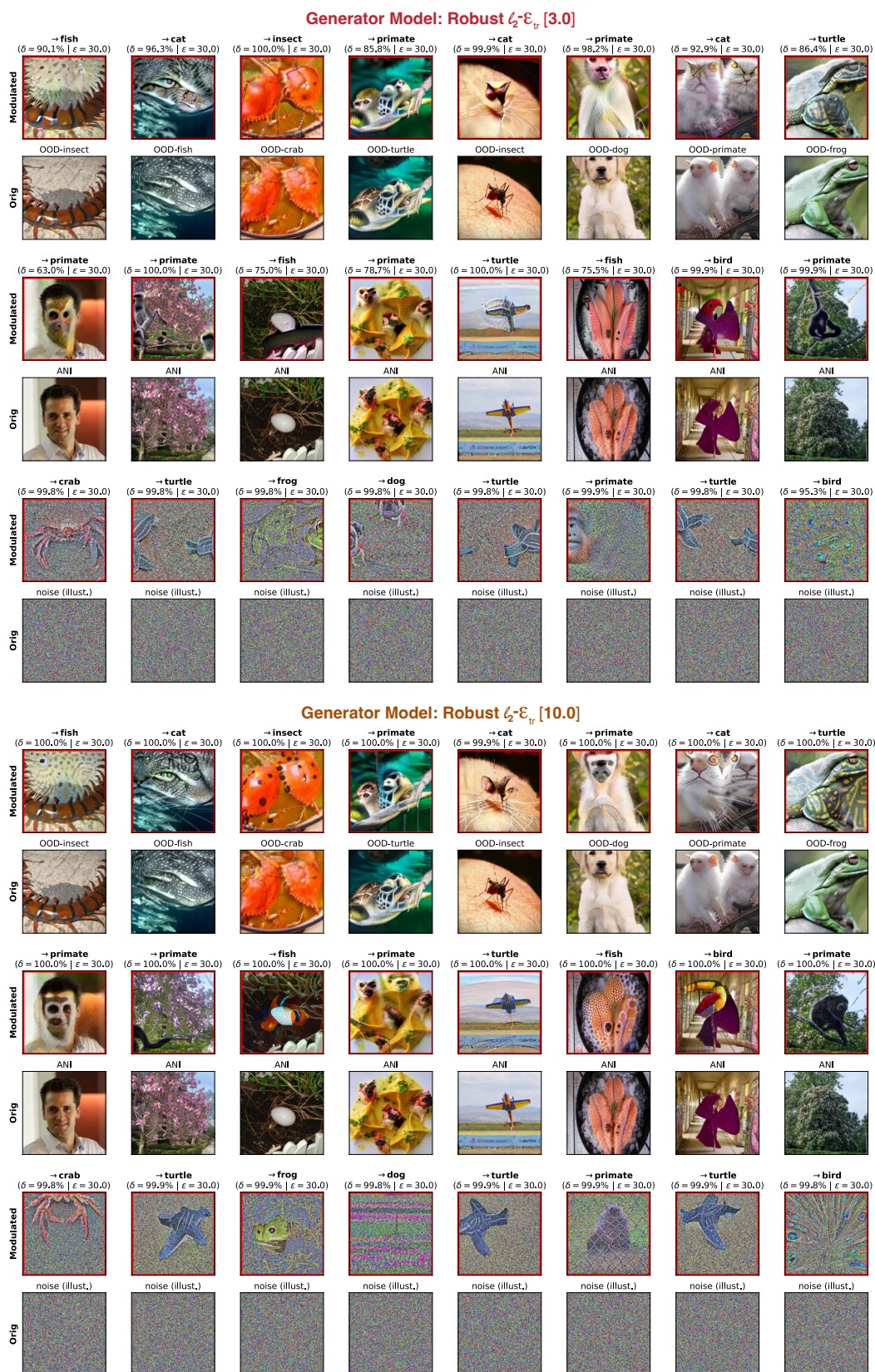

Figure 5: **Targeted Modulation examples using OOD, ANI, and UNI as source images.** *Perturbing at a pixel budget of 30.*

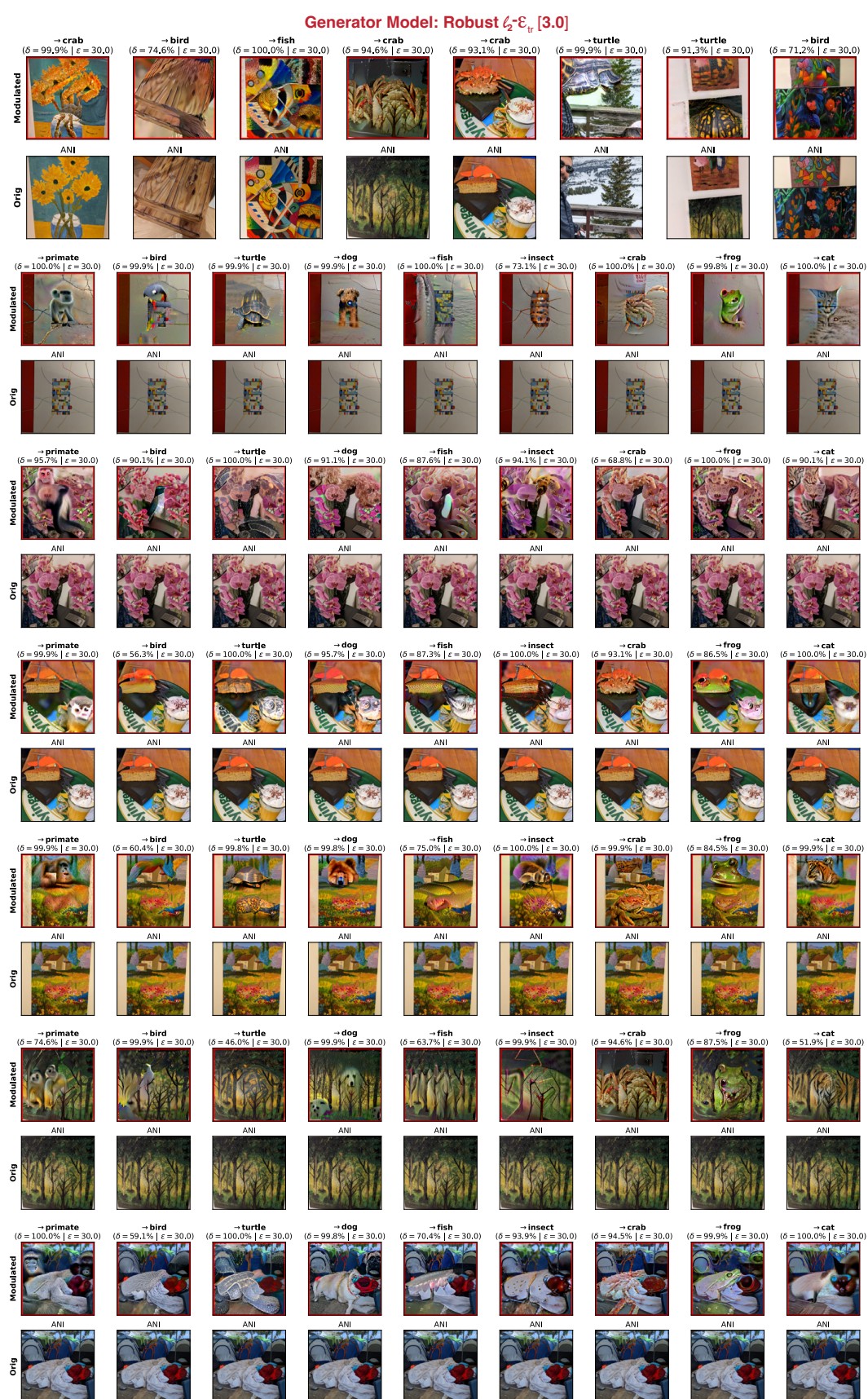

Figure 6: **Targeted Modulation examples using Arbitrary Natural Images (ANI) as source images.** *Perturbing at a pixel budget of 30. Source images were captured with a smartphone camera.*

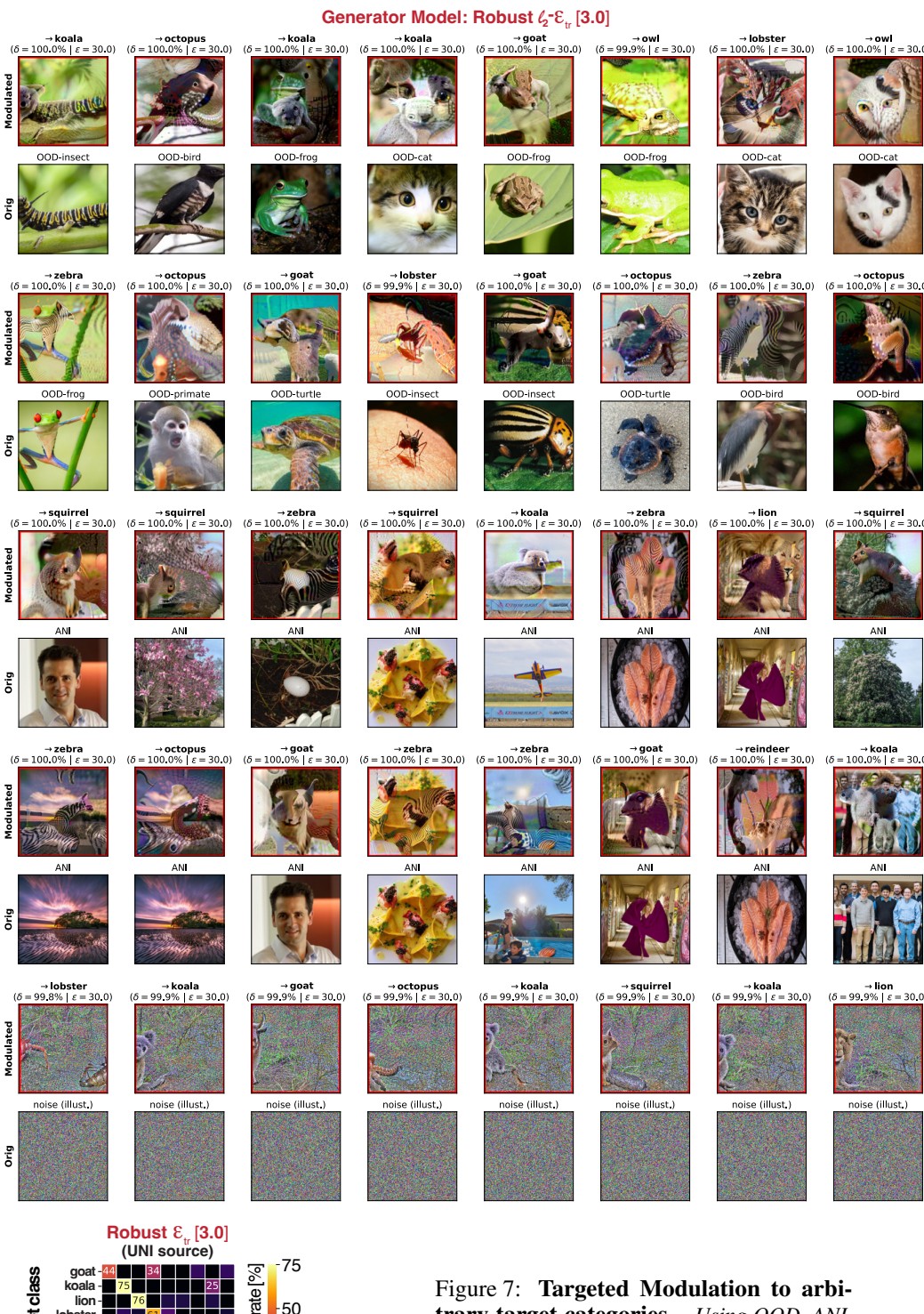

Figure 7: **Targeted Modulation to arbitrary target categories.** *Using OOD, ANI, and UNI as source images. Perturbing at a pixel budget of 30.*

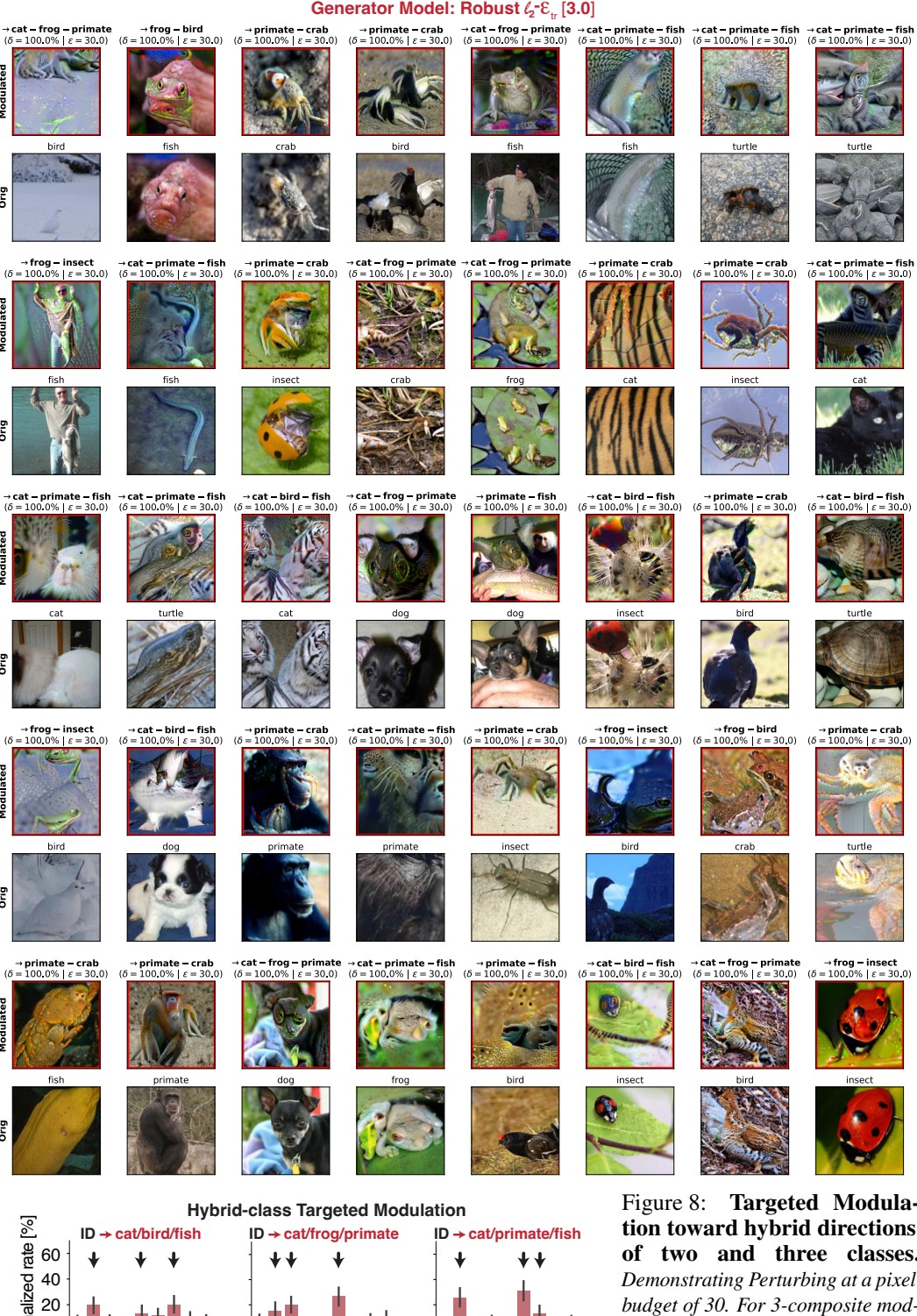

Figure 8: **Targeted Modulation toward hybrid directions of two and three classes.** *Demonstrating Perturbing at a pixel budget of 30. For 3-composite modulation a perfect success is ~33% rate for all three target categories.*

**Different attack strategy: $\ell_\infty$ attacks and $\ell_\infty$-robust models**

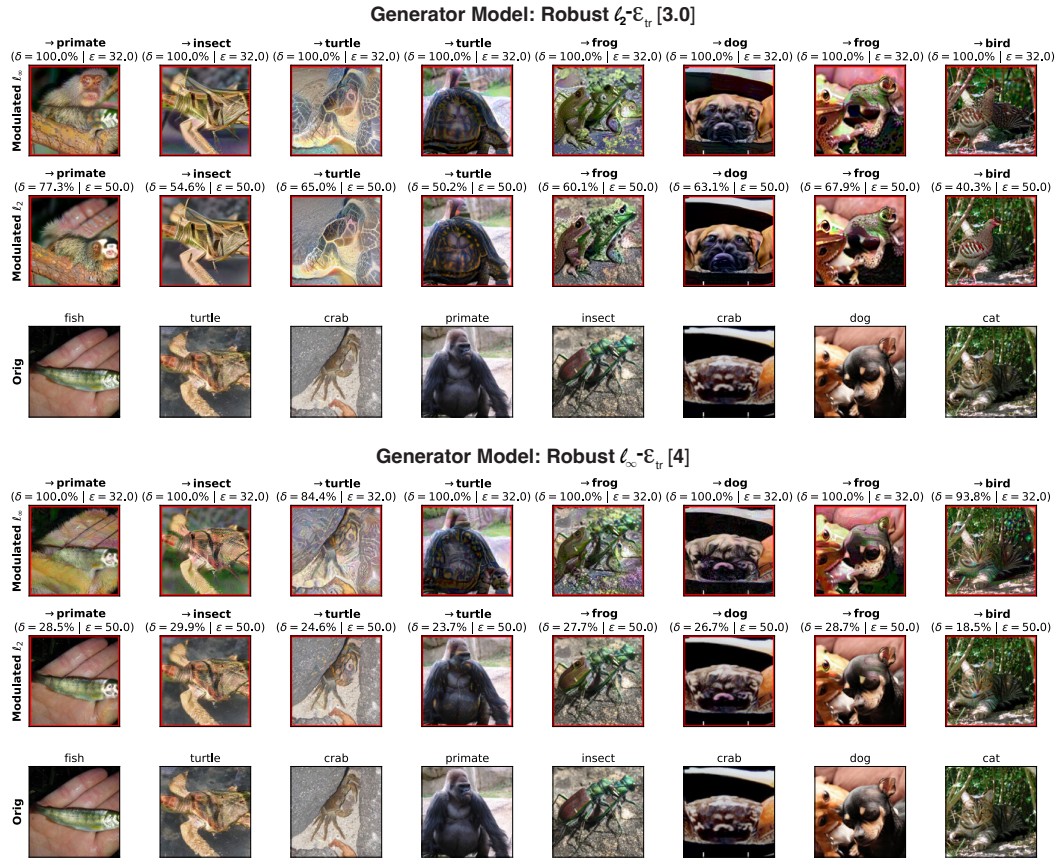

Figure 9: **Comparison with $\ell_\infty$ attacks and $\ell_\infty$-robust models.** *We present results for all four combination of model robustification strategy ($\ell_\infty$ or $\ell_2$), and attack strategy for image generation ($\ell_\infty$ or $\ell_2$). The actual pixel budgets for $\ell_\infty$ attacks (or models) are X/255 of those marked. We denote the proportion of pixel budget used for generating the modulated image from the source image by $\delta$. Bold titles above modulated images denote the targeted class label. Titles above original (source) images denote the ground-truth class label. We chose model/attack $\ell_\infty$ pixel budgets that are comparable with those used in the $\ell_2$ setting.*

**Restricted ImageNet class mapping**

In what follows we provide the definition of Restricted ImageNet classes by their corresponding ImageNet classes, as previously defined in robustness library [2].

● Dog (classes 151-268) ● Cat (classes 281-285) ● Frog (classes 30-32) ● Turtle (classes 33-37) ● Bird (classes 80-100) ● Monkey (classes 365-382) ● Fish (classes 389-397) ● Crab (classes 118-121) ● Insect (classes 300-319)