# OpenReview forum: "Strong and Precise Modulation of Human Percepts via Robustified ANNs"
_NeurIPS.cc/2023/Conference — NeurIPS 2023 poster_

### Official Review · Reviewer_ibQ6 · 2023-07-07

**Soundness:** 3 good
**Presentation:** 3 good
**Contribution:** 2 fair
**Rating:** 7
**Confidence:** 4

**Summary:**

The paper presents a novel approach to find categorical perceptual changes in humans using artificial neural networks (ANNs). Notably, the paper presents compelling evidence that an adversarially trained resnet50  is better at generating these adversarial attacks on humans on a low budget pixel regime. To test this the authors tested on humans subjects how the categorical percept change in relation with the pixel  budget assigned to a PGD attack on the network. On low pixel budget, adversarially robust  resnet50 are able to generate stimuli that can generate more successful targeted attacks that affect humans.

**Strengths:**

Overall the paper is well-written and  well-executed.

The question asked is interesting and very relevant to the community. Even though it’s not completely solved, the authors propose a first set of answers that could lead to interesting future work.

**Weaknesses:**

Minor points:

 I think having methods presented after the results was a bit unexpected and perhaps confusing. A reorganization may help understand a bit better the results, by putting the foundations first.
Be careful with Figure 2, the caption doesn’t exactly correspond to the figure : 3 panels in the figure but only two in the caption. Also in panel c, second row and “frog column", some writing appears which are probably not supposed to be part of the image.

the authors sometimes mentioned a pixel of 30 (or 30.0) and sometimes of 3.0. A revision to make sure the units are coherent through the paper would help.

The term “robustified models” can be a bit miss-leading and not commonly used. It would be better to refer to “adversely trained models”.


Major points:

Some of the contributions seem a bit  over-claimed considering that only one type of attack is tested and also  only one type of model is analyzed.

Although the pixel budget is well defined, there seems strong semantic distance even in low-pixel budget regime. For instance, in figure 1, the primate example, it seems like in 10^1 budget regime there is crucial semantic  information lost and added that would explain the impact on performance for humans. There seems to be potential in this direction to understand image classification, but the authors may need to clarify exactly why taking this route.

It seems what the paper is proposing is a categorical morpher, why taking this route and what does it tell us about the human judgements? Also, why using an adversarial approach and not a stable diffusion approach?

**Questions:**

Why are humans as sensitive to image perturbation at an l2 budget of 3.0 and 10.0? What happens with a pixel budget of 5 for example? And more than 10? How can you explain that by changing more content in the human, the images get less semantically different for humans?

How do you explain that DM images have more effects on human errors than TM perturbations?

It was not completely clear what is the performance of both models in the 9 categories described.

**Limitations:**

No explicit limitation section is written.
It would be nice to mention that this work is restricted to only one norm of adversarial attacks and one type of architecture. Those results show an increased alignment between humans and models’ robustness to adversarial attacks but a lot remains to be analyzed to completely claim of closing the gap.

---

> ### Author Rebuttal · Authors · 2023-08-03
>
> **Errors/typos & paper organization**
>
> We thank the reviewer for spotting those important errors/typos, now all corrected. As far as organization is concerned, and in accord with the reviewer’s suggestion, we designed the “Overview of approach and experiments” section to provide a succinct and high-level description of our methodology, while leaving mostly the more technical details for Methods. If there are additional elements, which if moved from Methods to Overview may help ease the reading further, we are happy to do so.
>
> **Budgets 30.0 and 3.0**
>
> Indeed, the 30.0 and 3.0 budgets are two distinct budgets, corresponding to the perturbation budget in our experiments when attacking *pre-trained models*, and that allowed *during model (adversarial) training*, respectively.  We also add a visual to explain these regimes to the global response figure and to Figs 2,3.
>
> We now further clarify this point by adding:
>
> In Overview:
>
> *“Notably, we denote this \textit{adversarial training budget} by $\varepsilon$, to be distinguished from the perturbation budget, $\epsilon$, which applies to any image-perturbation method.”*
>
> Kindly refer also to our global comment about this distinction.
>
> **The term “robustified models”**
>
> As correctly pointed out, our “robustified models” are colloquially known as “adversarially trained models” or “robust models”. However, adversarial training does not guarantee a robust model – not to adversarial attacks nor in broader senses of robustness. We thus believe that coining the term “robustified models” has a merit in referring to just the process performed on the model towards becoming more robust. We further use it interchangeably with “adversarially trained models”.
>
> **Limited attack/model types**
>
> We agree that analyzing other attacks and/or model types can better support our contributions. We thus include in our Supplementary Material additional analyses titled “Different attack strategy: linf attacks and linf-robust models”. We describe and refer to this section from the main text in line 228 under “Other pixel budget constraints”.
>
> We refactored our conclusion section to more explicitly call out the limitations (specifically, we focused on one architecture and mainly l2).
>
> **Inducing interpretable semantic shifts**
>
> As human observers ourselves, we agree with the reviewer (another human observer) that we also perceive a *strong semantic shift* even in the low-pixel budget regime.   Systematically testing and quantifying such measures of human perception relative to model “perception” is a key contribution of our work. The ability to induce those shifts in human reports in such a pixel budget regime – which is otherwise a "perceptually stable regime” as we demonstrated via random attacks experiments or a contrast blend approach – was not known to be possible before this study.
> Notably we have in fact conducted experiments also by a Stable Diffusion modulation approach. While those are outside the scope of this paper, they reflected that unlike robustified-models’ guided perturbations, compelling modulations by LDMs require *larger* budgets than our defined low-norm budget (i.e., < 30).
>
> **Why are humans as sensitive to image perturbation at an l2 budget of 3.0 and 10.0**
>
> Figs 2b or 3b show human reports for Disruption or Targeted Modulations respectively, for perturbation/attack budgets in range < 30, which are performed on an array of models, including vanilla and a few robustified models. Those robustified models were trained to various degrees of robustification levels, 1, 3, and 10. The curves on both Figs 2b or 3b for all robustified models 3, 10, and mostly 1 are monotonically increasing, indicating the humans are more strongly modulated by higher allowable budget perturbations. Kindly refer to our previous clarification (and the global) regarding distinguishing between the training and the experiment perturbation budget.
>
> **How do you explain that DM images have more effects on human errors than TM perturbations?**
>
> There are 8/9 (n categories=9) ways to disrupt a category but only 1/9 to induce a specific one, making the former an easier objective compared with the latter under a given pixel budget constraint.
>
> It was not completely clear what is the performance of both models in the 9 categories described.
>
> Please refer to our global response on the topic.
>
> **It would be nice to mention that this work is restricted to only one norm of adversarial attacks and one type of architecture. Those results show an increased alignment between humans and models’ robustness to adversarial attacks but a lot remains to be analyzed to completely claim of closing the gap.**
>
> As noted previously, we have shown that these results extend also to linf attacks and have analyzed linf models. Previous studies on model adversarial training suggest that the results obtainable by other architectures (e.g., ResNet101) may be comparable. We have further mentioned in Conclusion ln 339-343 that our findings are in a lower bound sense, and that there may be other models that are more behaviorally aligned:
>
> *“We emphasize that these are lower bound estimates because we are probing human vision with models that are imperfect approximations of the underlying neurobiology. Indeed, we do not claim that robustified ANN models discover all low-norm wormholes or that they never point to wormholes that do not actually exist. Notably, because we still see a residual gap in behavioral alignment (Figs 1c and 2b), other models, which are even more behaviorally aligned, must still exist.”*

---

> > ### Comment · Reviewer_ibQ6 · 2023-08-16
> >
> > I want to thank the authors for the rebuttal and for the experiments adding other strategies. I have decided to keep my current score.

---

### Official Review · Reviewer_StMT · 2023-07-07

**Soundness:** 4 excellent
**Presentation:** 4 excellent
**Contribution:** 3 good
**Rating:** 8
**Confidence:** 5

**Summary:**

I have read the rebuttal and will keep my (relatively high!) score as is.

It is a folk-theorem that human categorization behavior is robust to adversarial perturbations that are under an L2 norm of 30 or less. This paper shows that networks that have received adversarial training can be used to generate relatively low pixel-budget “adversarial” examples that are reliably mis-classified by humans under an L_2 norm of 30 or less. This closes one gap between human behavior and network behavior, in that the humans are disrupted to nearly the same extent that networks are.

**Strengths:**

+ the paper shows a novel result: That when adversarially-trained networks, which are a better match to primate neural representations than “vanilla” networks, the adversarial examples that they generate can disrupt human categorization with a much lower budget than is seen with vanilla networks.

+ the paper shows that these “robustified” networks can modulate relatively arbitrary images to relatively arbitrary targets with low budgets.

+ The paper is well-written and the experiments are thorough.

**Weaknesses:**

- Just a bit of terminology complaint: Generally, an adversarial example is by definition one that fools networks but not humans. So once humans perceive these as the target category, I would no longer call them adversarial.

- A bit of a philosophical complaint: The paper focuses on the fact that these low-budget examples fool humans and networks to conclude that now these networks are better models of the human visual system. I find that a bit distracting from the goal of discovering network properties that cause the network to be a better model of the human visual system. That is, humans don’t *need* adversarial training to not be fooled by adversarial examples. A more biologically plausible model would be one that shares this property with humans. I find adversarial training to be a kind of dumb response to the susceptibility of networks to adversarial examples - again, we should be looking for properties that make them not susceptible to these images in the first place - e.g., perhaps the right kind of recurrence, feedback, a capsule architecture (see below), or some other property will achieve this.

**Questions:**

lines 23-25: “However, individual ANNs are also notoriously susceptible to adversarial attack: the addition of tiny (e.g., small l2-norm) pixel perturbations to the model’s input, optimized to disrupt the model’s categorization of the original image [11–13].”
This sentence is awkward: one expects a verb in the clause after the colon, but it is just a noun phrase describing what an attack is. So, it’s ok, it’s just hard to parse - almost a garden path.

lines 36-39: for an example of a robust model that doesn’t require adversarial training, and causes computed adversarial examples to appear as the category they are trying to spoof, see https://arxiv.org/abs/2002.07405.

line 126: Fig 2a -> Fig. 2

line 160: Fig 2a -> Fig, 2b

Figure 4: I’m not clear whether these are confusion matrices or error matrices? I.e., are the entries the percent of times the network returns that answer, or are they the times that it doesn’t. The caption seems to mean the latter (1-hot). A confusion matrix usually shows the number or percent of times that the answer goes in that box, so if the network is doing well, the diagonal has high values.

line 184: do you mean precision in the technical sense (i.e., as in precision and recall)? If not, better to choose a different word.

**Limitations:**

The authors adequately discuss the possibly misuse of this technique.

---

> ### Author Rebuttal · Authors · 2023-08-03
>
> We thank the reviewer for supporting our work.
>
> **Terminology complaint**
>
> We agree that “adversarial image” is not well defined in the field and that the work we presented exposes the need for a clear definition of this phrase.  Our working definition – which is inline with the original methodological approach – has been that an adversarial perturbation for a model is an image perturbation that intends to cause the model to make a mistake (relative to a ground truth label). Strictly speaking, this definition is *not dependent* on the effect of that perturbation on human perception.  However, because the difference between model perception and human perception is what made adversarial examples most interesting to many, the reviewer’s assumed definition (different than ours) is understandable. To avoid this confusion, we propose to revise the manuscript accordingly:
>
> ln48: “adversarial sensitivity” -> “low-norm perturbation sensitivity”
>
> ln75: “low-norm adversarial sensitivity” -> “low-norm perturbation sensitivity”
>
> ln133: “adversarial sensitivity” -> “perturbation sensitivity”
>
> ln240: “adversarial perturbations” -> “model-guided perturbations”
>
> **Philosophical complaint**
>
> We agree that finding a model that is both predictive and biologically plausible in its developmental/learning trajectory is a major goal in the field. Indeed we are not claiming that adversarial training (AT) is the mechanism by which robustness in humans emerges. We rather consider AT as a method by which we arrive at an improved estimate of the adult state ventral stream and supported behavior, and a contribution of our paper is to demonstrate that.  Future work may consider alternative, more biologically plausible mechanisms that may give rise to a comparably aligned model in terms of predictivity (e.g., Topographic ANNs, Margalit et al. 2023).  We will add a line in the new “Limitations” section to make this clear.
>
> **Phrasing in lines 23-25**
>
> We see the point. Now changed to:
>
> “However, individual ANNs are also notoriously susceptible to \emph{adversarial attack}: Adding of a tiny amplitude (e.g., ultra-low $\ell_{2}$-norm) pixel perturbation to the model's input, which is optimized to disrupt the model's categorization of the original image.”
>
> **An example of a robust model that doesn’t require adversarial training and causes computed adversarial examples to appear as the category they are trying to spoof**
>
> We thank the reviewer for this reference. As mentioned in our comment about biological plausibility, we do not claim nor highlight adversarial training or the specifics of our algorithm for image generation given a robustified model, as key to behavioral modulation. For this we dedicated a conclusion paragraph in lines 339-343 clarifying:
>
> “We emphasize that these are lower bound estimates because we are probing human vision with models that are imperfect approximations of the underlying neurobiology. Indeed, we do not claim that robustified ANN models discover all low-norm wormholes or that they never point to wormholes that do not actually exist. Notably, because we still see a residual gap in behavioral alignment (Figs 1c and 2b), other models, which are even more behaviorally aligned, must still exist.”
>
> line 126: Fig 2a -> Fig. 2
>
> line 160: Fig 2a -> Fig, 2b
>
> **Fig 2 referencing typos**
>
> Thanks! Corrected.
>
> **Figure 4 matrices**
>
> Indeed confusion matrices, with high values on the diagonal for robustified models, but not for the control methods (Image interpolation, see Supplementary Material for the vanilla). The values shown are normalized rates for choosing the target class, same as in panels (a) and (c), and as denoted on the color bar.
>
> **line 184: do you mean precision in the technical sense (i.e., as in precision and recall)? If not, better to choose a different word.**
>
> We thank the reviewer for this suggestion. We now change: “precision” -> “specificity” in ln184, ln202.

---

> > ### Comment · Reviewer_StMT · 2023-08-16
> > **quick note about just one point**
> >
> >
> > I still don't like your revised sentence: “However, individual ANNs are also notoriously susceptible to \emph{adversarial attack}: Adding of a tiny amplitude (e.g., ultra-low-norm) pixel perturbation to the model's input, which is optimized to disrupt the model's categorization of the original image.”
> >
> > here's a suggested revision:
> >
> > However, individual ANNs are also notoriously susceptible to \emph{adversarial attack}: The addition of a tiny amplitude (e.g., ultra-low
> > -norm) pixel perturbation to the model's input that causes the model to mis-categorize the image.

---

> > ### Comment · Reviewer_StMT · 2023-08-16
> > **definition of adversarial attack**
> >
> > Hi - Again, we disagree about what an adversarial attack is, and I think your new terminology just obscures things. I asked a few of my friends, and they generally agree with my definition, that an adversarial attack is one that deceives the human. Geoff Hinton said, with respect to your wormholed-examples, where a human can see what it is: "I call that a deflected adversarial attack. In order to get the neural net to get it wrong, you had to change the image so much that a person also saw the image as something else." I.e., defining an adversarial attack as just making a perturbation that makes the model get it wrong is too broad a definition, and your examples are no longer adversarial - they are deflected adversarial attacks.

---

> > > ### Author Response · Authors · 2023-08-21
> > > **Definition of Adversarial Attacks**
> > >
> > > Thank you for the suggested definition of "adversarial attack", and the suggestion of "defected adversarial attacks".
> > > The precise definition of "adversarial attack" is not central to the contributions presented in the paper, and based on your comments we are now unsure of the consensus view of this term.
> > > As such, we are happy to drop the term completely or change it to align with the consensus terminology that may be reached among the reviewers and the area chairs.

---

> ### Comment · Reviewer_StMT · 2023-08-20
> **Why my score is high compared to the other reviewers'**
>
> First, let me say that I find the results reported in this paper to be (nearly) completely novel. The fact that in "robustified" networks, small budget changes make **semantically obvious** changes in the images is something I've never seen before. Hence I disagree completely that these results are not novel and to be expected (Reviewer keVx). This paper will attract a lot of attention because of this. (I say "nearly" because in my review I did cite a paper where a special capsule network that is attacked changed the appearance semantically  (deflected attacks) - but this was for MNIST digits - not nearly as interesting as this paper's results).
>
> Their first listed contribution: " We provide evidence for the existence of low-norm image perturbations of arbitrary natural images that strongly disrupt human categorization behavior." was clearly fulfilled - and I would argue the other contributions listed are also fulfilled.
>
> **Reviewer Xb7f (6: weak accept):** Their first weakness was "My main concern with the paper has to do with the interpretation. A perturbation size of 30 (for normalized images) seems quite large, and the example images do not (in my estimation) seem "close" to the original image."
> To my mind, this is one of the main points of the paper! A perturbation of size 30 for a vanilla network (that has not been adversarially trained) would **not** be noticeable. Here, it is - there is a semantic change that is striking in how it focuses on features that would change the classification, such as essentially adding a monkey face to the frog.
>
> And I completely agree with their interpretation that "Perturbations from adversarially trained networks are semantically meaningful,” which, if adopted by the authors, would be a nice way of putting their result.
> I don’t really see how this critique - how the results are interpreted	- makes the results less important.
>
> **Reviewer keV (4: borderline reject):** Main complaint: lacks novelty. As noted above, I completely disagree.
>
> **Review LMUo (5: borderline accept):** again, this reviewer says the result “doesn’t seem surprising.  Humans often misjudge many images”. See above.
>
> **Reviewer ibQ6 (5: borderline accept):** (small point: doesn’t like the term “robustified” - seems fine to me)
>
> Major complaints:
>
> - only one type of attack and one type of network tested. Response: You only need one flying pig to make a point. More seriously, the authors show the result also holds for l_{\inf} models.
>
> - Remarks that there is strong semantic distance, even in a low-budget regime. Again, to me, that’s the point of the paper - these are “wormholes” in representation space: a shortcut to another category.

---

### Official Review · Reviewer_LMUo · 2023-07-08

**Soundness:** 2 fair
**Presentation:** 3 good
**Contribution:** 2 fair
**Rating:** 5
**Confidence:** 3

**Summary:**

The paper systematically challenges the common assumption that human categorization of images remains highly robust to small-scale image perturbations (low pixel budget). The authors find that small-scale image perturbations, guided by adversarially trained artificial neural networks (robustified ANNs), can significantly and precisely alter human perception of object categories. Not only does this finding challenge the above assumption, but it also reveals the existence of "wormholes" in the image space that can lead to a significant change in human object category perception. Moreover, the study demonstrates that contemporary models of visual processing are precise enough to locate these "wormholes".

**Strengths:**

1) The authors discuss from an intriguing perspective how deep neural network models (robustified ANNs) can induce anticipated human behavior (i.e., misjudgments) with generated images (slight changes on the original image).
2) The authors reveal that only models generated through adversarial training have this ability to induce.
3) The authors seek to explore human perceptual wormholes from modern visual models (robustified ANNs).

**Weaknesses:**

1) The paper revolves around the premise that images generated by robustified ANNs (under a low pixel budget) can interfere with human classification judgments. This conclusion doesn't seem surprising. Humans often misjudge many images; it's just that robustified ANNs can generate highly deceptive images.
2) The authors believe that robustified ANNs and human perception have similar responses to image perturbations. The only evaluation criterion seems to be the final classification error rate. Having similar error rates doesn't equate to having similar responses.
3) The idea of using robustified ANNs to search for human perceptual wormholes seems impractical, since even from the perspective of robustified ANNs, the authors do not provide any patterns of wormholes in robustified ANNs.

**Questions:**

1) Have the authors tried more models besides ResNet?
2) Could the connection between deep models and human perception be explored from more angles, such as the similarity of intermediate features in the model?
3) Wormholes, a core part of the paper, are not prominently discussed. The authors should seek and describe the regularities of wormholes in robustified ANNs, instead of simply saying that they can generate images that interfere with human judgment from any source image distribution.

---

> ### Author Rebuttal · Authors · 2023-08-03
>
> **The paper revolves around the premise that images generated by robustified ANNs (under a low pixel budget) can interfere with human classification judgments. This conclusion doesn't seem surprising. Humans often misjudge many images; it's just that robustified ANNs can generate highly deceptive images**
>
> We thank the reviewer for this comment.
> We agree that attacks on robustified models were previously shown to engage human perception (as addressed in Introduction, ln38). However, to the best of our knowledge, there is yet no evidence for systematically quantifying the model-human gap in adversarial sensitivity. Further, it was unknown whether in a low budget-regime – in which, as we show, human perception is stable under baseline attacks (image interpolation or random attacks) – it is possible to disrupt or induce specific category percepts, and to what degree. Furthermore, and even more surprising, is that the strong modulations by robustified guide models are prominently evident when perturbing at budgets well beyond the budget range used for model robustification. Namely beyond the “robustification perimeter”.
>
> For perspective, consider some of the following live hypotheses prior to this study: Image perturbations discovered by robustified ANNs (1) have no effect on human category reports, regardless of the perturbation budget, (2) have some effect of on human category reports, but only in the typical-norm budget regime (i.e. typical of pairwise distances, ~130), (3) have perceptually interpretable effects in the low-norm regime, but have no effect on human categorization behavior.
>
> To better clarify our results and their novelty we now revise in Results:
>
>
> *“Notably, we report these strong disruptions in human category percepts when perturbing at budgets well above the budget used for model robustification, yet still well-below the typical pairwise natural image distance regime. This result was not a priori guaranteed.”*
>
> **Have the authors tried more models besides ResNet?**
>
> While generalizing to other architectures is not foundational to the our claims presented in this paper, we agree that exploring other architectures is an interesting question.
>
> Because model adversarial training takes ~8 days to complete (on 4 A100 GPUs), in conjunction with the requirement to train two such models per architecture and robustification level for white-box/gray-box analysis, for this study, we focused our analysis on ResNet50 only.
>
> Previous studies on model adversarial training suggest that the results obtainable by other architectures (e.g., ResNet101) may be comparable (Singh, Croce & Hain 2023). We have further mentioned in Conclusion ln 339-343 that our findings are in a lower bound sense, and that there may be other models that are more behaviorally aligned:
>
> *“We emphasize that these are lower bound estimates because we are probing human vision with models that are imperfect approximations of the underlying neurobiology. Indeed, we do not claim that robustified ANN models discover all low-norm wormholes or that they never point to wormholes that do not actually exist. Notably, because we still see a residual gap in behavioral alignment (Figs 1c and 2b), other models, which are even more behaviorally aligned, must still exist.”*
>
>
> **Model-human alignment evaluation criterion focusing on final classification**
>
> We agree with the reviewer that finding a good model of biological vision has multiple facets to it, including internal representation. We note that while our disruption modulation (DM) experiments were indeed simply measures of error rate (accuracy), our Targeted Modulation (TM) tests were tests of specific types of errors, which is a more stringent test in the spirit of the reviewer’s question. Internal (feature) similarities of these models and primate internal neural representations have been reported prior to this study, and were key motivation points for us to focus the present paper on testing the alignment in the downstream behavioral output, which is conceived as supported by those internal responses in both the brain and the models. We refer to those prior findings about the similarities in representation between the biological visual system and the artificial counterpart in the Introduction in lines 39-46.
>
> **Using robustified ANNs to search for human perceptual wormholes**
>
> We thank the reviewer for raising this concern, which understandably can be elusive. If correctly parsed, this comment has to do with better explaining what we mean by “wormholes”. The “wormholes” is a metaphor for our empirical finding that, from every tested starting “location” (i.e., starting image), there exists a “nearby” (i.e., low-norm) perturbation step that will induce drastic changes in human behavioral categorization report. The metaphor is intended to conceptually express the idea that when a human subject is in one perceptual “universe”, s/he can “walk” a very short distance in pixel space to then be in another perceptual “universe.” We acknowledge that this metaphor may not be perfect, and may be missed, but we are inclined to believe this helps convey the conceptual essence of our findings in a faithful way.  We will add a sentence in Introduction to clarify this.
>
> We agree with the reviewer’s related comment that we have not yet fully characterized the statistical distribution of such human perceptual wormholes at any given pixel distance, but we believe the work we presented here – demonstrating that some computational models can reliably point to and reveal those human perceptual phenomena – is a necessary precursor for that follow on work.  We have added a sentence in the revised manuscript (under “Limitations”) to express this.

---

### Official Review · Reviewer_keVx · 2023-07-10

**Soundness:** 4 excellent
**Presentation:** 4 excellent
**Contribution:** 2 fair
**Rating:** 4
**Confidence:** 5

**Summary:**

The paper under review brings to light a crucial and fascinating issue in deep learning: adversarial attacks. The authors contend that a neural network trained adversarially, which is logically more robust against attacks, produces perturbed images that humans perceive as differing from the original when it is finally attacked. They present evidence that humans perceive images generated for both non-targeted and targeted attacks similar to how the neural network does. The concept of comparing human perception with the performance of artificial neural networks holds merit. However, it's necessary to critique the exaggeration of novelty in the authors' findings. They find that an adversarially trained network, which necessitates higher pixel perturbations for successful attacks due to its robustness, aligns well with human perception. Further, the authors' claim of their 'novel finding' that "human percepts are massively disrupted by low-norm image perturbations discovered by robustified ANNs”. The purported surprise and novelty is  questionable as it simply reiterates the foundational goal of adversarial training and warrants scrutiny. Firstly, the paper does not offer a clear foundation as to who assumes that humans are necessarily resilient all perturbation with low norm, casting doubt on the originality of the claim. Secondly, as the authors themselves acknowledge (figure 1)  that object classes are intricately entangled in pixel space, hinting that in high dimensions small pixel-space deviations can be found to cause perceptual shifts to other classes. Therefore, it's counter-intuitive to assert that such a phenomenon is a surprising novelty. Moreover, the paper employs a pixel budget of 30 to define 'small perturbations', but it falls short of explaining why this particular value is low (average and maximum distances in the ImageNet dataset's high-dimensional space is 130 and 338 pixels if these distances are to be compared to the pixel budget for adversarial generation). To conclude, despite reservations regarding the claimed novelty, the paper does contributes valuable insights. Its methodology is solid, and its attention to human perceptual measurements is commendable. The authors carry out a diligent study quantifying human perception with respect to adversarial attacks, the results of which could spark important discussions in the field of adversarial robustness. However, the paper lacks claimed novelty.

**Strengths:**

Quantification of human perception relating it to adversarially trained neural networks.

**Weaknesses:**

-Lacks novelty and main results are expected based on the fact that the whole purposed that neural networks are trained with adversarial attacks is to make robust and more aligned with human perception
- Not clear why in Figure 1c1 there is not shift of the red curve to the left compared to the blue one which is what one would expect from robustified neural net. The authors show this expected shift in Figure 2b where the blue is clearly shift to the left.

**Questions:**

Explain better the value of a pixel budget perturbation (i.e. 30). For example you can compare to contrast of the image to provide a more intuitive meaning of the level of perturbation needed to change classes for different networks.

**Limitations:**

Yes

---

> ### Author Rebuttal · Authors · 2023-08-07
>
> We thank the reviewer for reviewing our work, for their positive comments about our behavioral experimental methods, and for their critical analysis of our empirical findings. We agree that the novelty and significance of our findings are dependent on 1) the prior beliefs in the field on perceptual sensitivity in humans, 2) the extent to which our results are straightforward consequences or restatements of adversarial training, and 3) whether any natural image may, with high probability, be a distance of < 30 to other images from other categories in pixel space (which would make it wholly unsurprising that perceptual shifts via perturbations of norm < 30 are possible ).
>
> We provide our clarifications and justification on each of these points below.
>
> **Who assumes that humans are necessarily resilient to all perturbations of low norm?**
>
> We contend that human vision is widely believed to be robust to “ultra low norm” perturbations (l2 norm < 3); e.g. [Wichman and Geirhos, 2023](https://arxiv.org/pdf/2305.17023.pdf) and [Dujmovic et al. 2020](https://elifesciences.org/articles/55978#s3).
>
> However, we should have been careful to not make the logical leap that this implies that the field also widely believes humans are invariant to merely “low norm” perturbations (which we define as norms < 30; see our global response for justification), where we found evidence of human perceptual sensitivity. We implicitly make this leap throughout our paper (e.g. language in line 30; line 323), and intend to be more precise in a future revision. Indeed, it would be more accurate for us to describe our findings not as challenging a “prevailing assumption”, but rather as resolving wide uncertainty and an absence of evidence around the question of whether and to what extent humans are sensitive in this < 30 “low norm” regime. We are unaware of any other prior work systematically establishing the perturbative sensitivity of human perception in this norm regime (and would welcome any key references in this regard).
>
> We thank the reviewer for challenging us on this point, which will help us more accurately contextualize our findings relative to existing beliefs.
>
> **Is alignment with human perceptual sensitivity to “small” pixel perturbations a restatement of adversarial training?**
>
> The adversarial training (AT) objective is to make a system’s behavior robust to pixel perturbations less than or equal to a certain norm (which is determined by a hyperparameter). This objective could be aligned with the objective of driving similarity to human perceptual sensitivity, to the extent that humans are **also** robust to perturbations below that hyperparameter-specified norm.
>
> However, our results focus on the fact that we found correspondences in sensitivity between AT-trained guide models and human perception at perturbation budgets **outside** of the budget range used for AT. This result could not be a straightforward consequence of the AT objective, which provides no explicit constraints on perturbations greater than 3 (for the primary, epsilon=3 robust model we considered).  Indeed, the degree of alignment between humans and models has not been previously shown or systematically tested and analyzed prior to this study.
>
> **To what extent are images from different object classes “close” to each other in pixel space?**
>
> As shown in Rebuttal Panel A, and as the reviewer mentioned in their answer, natural images are estimated to be typically farther than what we termed “low norm” (median pairwise l2 distance ~130; 99.9% quantile range [69, 268], min ~45; max ~306; from Imagenet Restricted images).
>
> As the reviewer correctly points out, if we had found that humans were perceptually sensitive from perturbations with norms of “typical” size, this would have been a trivial and unsurprising result (in that humans are obviously perceptually sensitive to different natural images).
>
> However, we found humans were perceptually sensitive to perturbations with norms as low as ~10. This is roughly one order of magnitude lower than the trivial result.
>
> **Our response to: “Not clear why in Figure 1c1 there is not shift of the red curve to the left compared to the blue one which is what one would expect from robustified neural net. The authors show this expected shift in Figure 2b where the blue is clearly shift to the left.”**
>
> Thank you for noticing this. Some context: you are absolutely correct that a robustified version of a model should be more invariant to perturbations, namely the red curve should be to the right of the blue curve (i.e. requires a higher pixel budget to produce the same level of disruption). We indeed replicate that previously reported result in Fig 2a (left panel). However,  please note that this rightward shift is only guaranteed in the situation in which the model is used to design attacks against itself (aka “white box”).  The colored curves in Fig 1c1 are the results of “gray box” attacks. Specifically, these are tests of attacks designed by one member of the model family and tested on another member of that same model family. We plot these against the human data in Figure 1 as they are the proper way to make a quantitative comparison to the behavioral of human visual neural networks (as we do not have white box access to those networks), and we explain this in Overview. For gray box comparisons, the rightward shift of the red curve relative to the blue curve is, unlike the white-box situation, not guaranteed.  Indeed, while not the primary focus of our work, it is interesting that we found that robust models and vanilla models had similar gray box sensitivities (e.g. as shown in Figure 1 and Fig 2b; left panel).  We will add a clarification to the caption of Fig 1 to indicate that these are gray-box results

---

> > ### Comment · Reviewer_keVx · 2023-08-18
> >
> > I appreciate the authors' attempt to refine and temper their conclusions. Nevertheless, I continue to assert the significance of discussing the distances between natural images more extensively in the paper. Consider this: if a random search across millions of natural images reveals that there is even one image whose distance from another image image to change its class falls within tens of pixels (in Euclidean distance), then even if the authors' method can synthesize such images, the outcome is not particularly surprising. The authors indicate (median pairwise l2 distance ~130; 99.9% quantile range [69, 268], min ~45; max ~306; from ImageNet Restricted images), with a minimum of ~45, which closely aligns with their values. This essentially reinforces my initial argument. Therefore if one can identify even a single image from millions that has a tens-of-pixels distance, and their method yields a result within the same magnitude, then this should be clarified and quantified. The paper should be written with this emphasis in mind.

---

> > > ### Author Response · Authors · 2023-08-21
> > >
> > > We agree with the reviewers intuition that wormholes may be a naturally occurring phenomenon, even amongst natural images. Finding a natural image that falls within the 30 pixel budget and disrupts the category might be possible, especially for carefully chosen start images.
> > > However, the number of those candidate images is unfathomably large (well beyond millions).
> > > Our point is that there exist models that allow us to find perceptual wormholes very efficiently near arbitrary start images. This is not to claim that "natural wormholes", as given by a search over an infinite natural image database, is impossible.
> > > However, we don't believe such an efficient search has been demonstrated before nor has been shown to approach budget-limited disruption rates as those shown in this paper. We are happy to properly address such relevant past literature if exists.

---

### Official Review · Reviewer_Xb7f · 2023-07-11

**Soundness:** 3 good
**Presentation:** 4 excellent
**Contribution:** 3 good
**Rating:** 6
**Confidence:** 3

**Summary:**

This paper studies the problem of adversarial images in artificial and biological visual systems. On standard ("vanilla") image models (e.g. ResNet-50), adversarial perturbations that successfully disrupt a vanilla model are quite small and do not significantly alter human perception. This paper generates adversarial perturbations on adversarially-trained ("robustified") image models, and tests whether those perturbations more effectively disrupt human perception, and find that they do (especially as the perturbation norm grows larger). The paper finds that it is both possible to perturb in an arbitrary direction, or in a targeted direction towards another class. It appears that you need to adversarially train the network used to generate the perturbation with a strong perturbation strength in order to see these effects (networks trained with no or weak adversarial perturbations did not yield perturbations that generalized to humans).

**Strengths:**

- Experiments are clear and straight-forward
- Effect size is significant, although the perturbation size required is large.
- Experiments contain nice controls and baselines (comparing perturbations from four different adversarially trained networks trained with different perturbation strengths; comparing targeted perturbations to interpolation; hybrid-targeted modulation, etc.)


**Weaknesses:**

- My main concern with the paper has to do with the interpretation. A perturbation size of 30 (for normalized images) seems quite large, and the example images do not (in my estimation) seem "close" to the original image. The original definition of adversarial images were images that were quite close in pixel space (imperceptibly so); by contrast the perturbations needed to modulate human perception are quite large. I don't think that in and of itself is an issue, it's still interesting that the large perturbations from an adversarially-trained network are sufficient to modulate human perception (especially in a targeted way). What I disagree with is that the claim that these perturbations are "tiny" (e.g. first sentence of the abstract, Fig 1 caption, etc). Throughout the text, the paper suggests that these are really small perturbations, which doesn't feel correct to me at a norm of 30.
- Instead, my interpretation of the paper would be something like: "Perturbations from adversarially trained networks are semantically meaningful". If you look at some of the example perturbations, they are intuitive (unlike adversarial perturbations on vanilla networks, which look like noise). For example, in Fig 1c2, to perturb the image of a frog, the perturbation simply replaces the frog's head with that of a primate, and replaces the frog's skin texture with fur. By keeping the same background color (black) and pose of the animal, the perturbation minimizes l2 norm in pixel space while still being semantically meaningful. In some sense, the perturbation has become more _efficient_. Rather than a random noise pattern fooling the network, the perturbation needs to be more careful in how it modifies the target image. It is no surprise then that these more effective perturbations also modify human perception. I don't think this interpretation is any less interesting, but to me it feels like a more accurate description of the results in the paper.
- I am curious if the authors looked at similarities or differences in the representations across the four trained networks. If one expects the robustified networks to be more similar to biological representations, this could make predictions for experimental visual neuroscientists to test.

**Questions:**

- I would appreciate more details on the four adversarially trained networks. What is their performance on restricted imagenet? What do the training curves look like?
- I would also like to get some better intuition for what a pixel norm of 30 means in this context.

**Limitations:**

Yes

---

> ### Author Rebuttal · Authors · 2023-08-03
>
> **Interpretation of the low budget regime**
>
> We thank the reviewer for this feedback. Please refer to our global response on this point.
>
> **I am curious if the authors looked at similarities or differences in the representations across the four trained networks. If one expects the robustified networks to be more similar to biological representations, this could make predictions for experimental visual neuroscientists to test.**
>
> We thank the reviewer for this suggestion. Some connections between adversarial robustness and increased representation alignment between models and primate neural activity have been established in previous works ([Dapello et al. NeurIPS’20](https://proceedings.neurips.cc/paper/2020/hash/98b17f068d5d9b7668e19fb8ae470841-Abstract.html), [Guo et al. ICLR’22](https://proceedings.mlr.press/v162/guo22d/guo22d.pdf), [Dapello et al. ICLR’23](https://openreview.net/pdf?id=SMYdcXjJh1q)), as we discuss in lines 36-46. Further evaluating the correspondences between the model(s) we study in this work and biological representations is an interesting line of future work.
>
> **I would appreciate more details on the four adversarially trained networks. What is their performance on restricted imagenet? What do the training curves look like?**
>
> Please refer to our global response on the topic.
>
> **I would also like to get some better intuition for what a pixel norm of 30 means in this context.**
>
> Please refer to our global response on the pixel norm value.  Notably most of the studied low pixel budget range (i.e., < 30) is beyond the robustification budget range (1.0, 3.0, or 10.0). In this context, please also refer to our global response on distinguishing these two budgets (adversarial training pixel budget vs. the budget used in our behavior modulation experiments on the pretrained models).
>
> To better illustrate the meaning of a 30 pixel budget, we show examples of pixel-norm 30 perturbations (alongside others) Rebuttal Panel B for a variety of guide approaches.

---

> > ### Comment · Reviewer_Xb7f · 2023-08-19
> > **Thanks for your response**
> >
> > I am still curious what the authors' thoughts are on my interpretation of the paper ("perturbations from adversarially trained networks are semantically meaningful"). I'm not suggesting you make drastic changes to the paper, but curious if you agree with my interpretation.
> >
> > After reading the other reviews and the rebuttal, I have decided to keep my original score.

---

> > > ### Author Response · Authors · 2023-08-21
> > >
> > > We tried to adhere to the signal measured, in this case the category report. But if we assume that "semantically meaningful" is defined as that which causes a category report change, then yes, we agree with this interpretation.

---

### Author Rebuttal · Authors · 2023-08-03

We thank the reviewers for finding our work interesting and for your support. We appreciate your constructive feedback and we agree that those suggestions would clarify the contributions of this work.

**Interpreting the low-pixel budget regime (< 30)**

All reviewers asked for clarification about the “low-norm” pixel budget regime that we focused on in the paper.  We agree that the original version of the paper did not provide full clarity on this issue.
Our definition of the “low-norm” perturbation pixel budget regime (l2 norm of < 30) was/is primarily based on empirical evidence that we provided in Figure 1 in our Supplementary Material. We show this empirical evidence in Rebuttal Panel A, and it illustrates that: 1) that human reports are virtually unaffected by random noise image perturbations or to image perturbations guided by a vanilla ANN, when those perturbations are restricted to pixel budgets less than 30, and 2)  that the typical (median) pairwise distance in natural images is ~130 with the *closest observed* pair of natural images (out of n=101,025 pairs) is ~45 apart. These empirical observations support the notion that perturbations of norm less than 30 can be reasonably referred to as “low-norm.”

As shown in Rebuttal Panel A, we define our perturbation budget regimes as:

- __“Ultra low-norm”__: < 3, a typical range studied in adversarial literature (e.g., for performing adversarial training)
- __“Low-norm”__: < 30, a range in which human categorization reports are insensitive to attacks by Gaussian noise perturbations or vanilla ANN attacks.
- __“Typical-norm”__ :  69-268, the range that we estimated contains 99.9% of pairwise distances for two independently sampled natural images.


In our revision, we will strictly adhere to these terms, and as suggested, we will refrain from calling the perturbations in our experiments “tiny” and simply call them “low-norm”.

To better illustrate the state of affairs prior to this study, we now also include in Rebuttal Panel B examples of pixel-norm 30 perturbations (alongside others) for three baseline image perturbation generation approaches.

As correctly pointed out by several reviewers, the perturbations discovered by robustified models – despite being low-norm – are indeed semantically meaningful, which is another way of saying that they are likely to produce specific changes in human perceptual report, and quantifying that was the focus of our study.  The primary novelty is the observation that, from apparently any starting image, robust guide models can be used to induce strong shifts in human perceptual reports in such a low-norm budget regime – otherwise a perceptually stable regime under naive baseline approaches.  This finding is novel in that it was not a priori guaranteed or otherwise known (see below).

**Distinguishing between pixel budget for adversarial training and pixel budget for designing perturbations experiments to test on humans and (fixed) models**

To further clarify the distinction between these two pixel budget regimes and their relationship to our results we now add in the revised manuscript:

In Overview:

*“Notably, we denote this \textit{adversarial training budget} by $\varepsilon$, to be distinguished from the perturbation budget, $\epsilon$, which applies to any image-perturbation method.”*

Please note that the adversarial training procedure used to train the robust guide models does not a priori guarantee that the resultant representations of robust models will be aligned with human perceptual semantics – rather, that procedure attempts to establish *complete insensitivity* to all pixel perturbations below a certain (ultra low) norm (determined by the epsilon-training hyperparameter), without any explicit constraint attempting to explicitly match measurements of human perception.  Thus, the fact these guide models automatically discover human-aligned perturbations in the (untrained) low-norm regime was not guaranteed, and is thus, in our view, surprising and novel.

To clarify this, we now add in Results:

*“Notably, we report these strong disruptions in human category percepts when perturbing at budgets well above the budget used for model robustification, yet still well-below the typical pairwise natural image distance regime (see Supplementary Material, Fig~1).  This result was not a priori guaranteed.”*

We also add a visual to explain these regimes to the global response figure and to Figs 2,3.


**Details on model performance**

We thank the reviewers for this feedback.
Given that these are ImageNet trained models they were evaluated on the 1000-way task on natural images and adversarial images (and for various evaluation budgets for the adversarial attacks). Their performance on natural images ranged from 45-76% Top-1 accuracy with the least robustified (vanilla) highest; on adversarial images ranged from 10-53% Top-1 accuracy (when evaluating at the same budget as the training one) with the least robustified (excluding the vanilla) highest. As shown in prior work, robustification tends to reduce natural image accuracy (Tsipras et al. 2019, Zhang et al. 2019). Further note that robustifying at a higher training budget level entails evaluation at the same (higher) budget level, resulting in a lower validation accuracy on adversarial images overall for the models trained for higher robustification levels.

For reference we include in our response figures summarizing their performance and learning curves.

**Fig2 typos**

We thank the reviewers (StMT, ibQ6) for spotting Fig 2 typos, which are now all fixed.


**Public availability of behavioral data**:
If this work is accepted, we will release an anonymized copy of the behavioral data used in this study, following the regulations for privacy, anonymity, and storage in our protocol, which was approved by our institutional board. We will update the Methods section paper with a reference to this protocol, and a link to these data.

---

### Decision · Program_Chairs · 2023-09-21

**Decision:**

Accept (poster)

**Comment:**

In this paper, the authors explore the problem of adversarial images in both artificial and human visual systems. Typically, adversarial perturbations applied to convolutional neural networks (e.g. ResNet-50) may be small and imperceptible to humans, but may profoundly influence the predictions of the CNN. In this work, the authors demonstrate that adversarial perturbations targeted to an adversarially trained network (i.e. 'robustified') may generate perturbations that are semantically meaningful to a human. This effect is quite noticeable especially as the norm of the perturbation is relaxed. Furthermore, the authors find that they perturbations may be targeted to selectively change the classification of the object to arbitrary objects of choice.

The reviewers commented positively on the quality of the results, the semantic saliency of the resulting images (i.e. strong effect), and the clear and straight-forward experiments. The reviewers also surfaced concerns about the interpretation of the results. In particular, the pixel budget of 30 permitted in the adversarial attack seemed quite large and might confuse the original definition of adversarial images (e.g. Szegedy et al, 2013; Goodfellow et al, 2014).

That said, all reviewers were impressed by the visual saliency of the results but clearly there are some unresolved concerns about how to interpret these results. I found the suggestion by one reviewer (and concurred by a second reviewer) that one way to clarify the interpretation of the paper is to potentially retitle the contribution to be "Perturbations from adversarially trained networks are semantically meaningful". This would add far more clarity than introducing esoteric and unnecessary nomenclature such as "wormholes" and would bring clarity to the presentation. Regardless, these results are compelling and the larger community would benefit from their publication. For these reasons, this work will be conditionally accepted so long as the title and main description of the paper better align with the proposed phrasing above.